# Peptide clustering enhances large-scale analyses and reveals proteolytic signatures in mass spectrometry data

Erik Hartman [1] ✉, Fredrik Forsberg[2], Sven Kjellström [3], Jitka Petrlova [2], Congyu Luo [2], Aaron Scott [1], Manoj Puthia [2], Johan Malmström [1,4] & Artur Schmidtchen [2,4]

Recent advances in mass spectrometry-based peptidomics have catalyzed the identification and quantification of thousands of endogenous peptides across diverse biological systems. However, the vast peptidomic landscape generated by proteolytic processing poses several challenges for downstream analyses and limits the comparability of clinical samples. Here, we present an algorithm that aggregates peptides into peptide clusters, reducing the dimensionality of peptidomics data, improving the definition of protease cut sites, enhancing inter-sample comparability, and enabling the implementation of large-scale data analysis methods akin to those employed in other omics fields. We showcase the algorithm by performing large-scale quantitative analysis of wound fluid peptidomes of highly defined porcine wound infections and human clinical non-healing wounds. This revealed signature phenotype-specific peptide regions and proteolytic activity at the earliest stages of bacterial colonization. We validated the method on the urinary peptidome of type 1 diabetics which revealed potential subgroups and improved classification accuracy.

Peptides are short sequences of amino acids that, like proteins, play diverse and crucial roles in many biological processes. Recent advances in mass spectrometry instrumentation, sample preparation techniques, and data analysis strategies have catalyzed the large-scale studies of peptidomes[1,2]. While a fraction of endogenous peptides is synthesized de novo, a vast peptidomic landscape is generated through the degradation of proteins by endo- and exoproteases[3,4]. The interplay between these classes of proteases results in clusters of peptides with overlapping sequences centered around endoprotease cut sites, introducing variation and redundancy into peptidomic data. The large number of potential peptides combined with the stochastic sampling associated with data-dependent acquisition mass spectrometry makes peptidomes highly dynamic, typically resulting in diverse peptidomes with a low degree of overlap between samples and a large portion of missing values.

Current peptidomic analysis strategies largely rely on computational methods to sift through large datasets to identify relevant bioactive peptides[5–7]. This can be accomplished by, e.g., predicting the functions and properties of peptides and thereafter selecting the highest-scoring peptides according to a desired metric. While this 'needle-in-a-haystack' perspective has proven successful in identifying endogenous bioactive peptides and peptide biomarkers across diverse contexts such as diabetes, organ failure, inflammation, infection, cancer, neurodegenerative disease, and as neuronal peptide hormones[2,5,6,8–12], it entails disregarding a substantial fraction of the peptidome, potentially discarding important information about the biological system in question.

Wound infections represent a complex biological system where peptides play an important role, as bacteria release proteases and

[1]Division of Infection Medicine, Department of Clinical Sciences Lund, Faculty of Medicine, Lund University, Lund, Sweden. [2]Division of Dermatology and Venereology, Department of Clinical Sciences, Lund University, Lund, Sweden. [3]Division of Mass Spectrometry, Department of Clinical Sciences, Lund University, Lund, Sweden. [4]These authors contributed equally: Johan Malmström, Artur Schmidtchen. ✉e-mail: erik.hartman@med.lu.se

other factors that modulate the hosts' proteolytic activity to shape the peptidomic landscape, facilitate immune response subversion, and promote bacterial invasion[4,13–17]. Concurrently, the host has developed elaborate peptide-based defense systems that aid in combating the pathogen as a part of the innate defense system[18,19]. The peptidome lies at the interface of this interaction as it reflects the combination of substrate availability, protease landscape, protease activity, post-translational modifications, and conformally exposed cut sites. These may vary depending on the microenvironment and type of pathogen underscoring the need to investigate protein degradation and the resulting peptidome for mechanistic insights and as a source for both biomarkers and therapeutic targets for wound infections[4]. Large-scale peptidomic studies could potentially provide a more comprehensive perspective on these mechanisms, however, there remains an unmet need for computational methods that enable large and unbiased quantification of peptidomes.

This is especially relevant as wound infections constitute a substantial societal burden due to their impact on public health and healthcare resources, the challenges involved in their diagnosis, and the pathogens' resilience to treatments. Several bacterial species, such as *Pseudomonas aeruginosa*, *Escherichia coli*, *Acinetobacter baumanii*, *Corynebacterium*, *Enterococcus faecalis*, and *Staphylococcus aureus* are particularly common in wound infections. Of these, *P. aeruginosa* and *S. aureus* are the most common culprits in burn wounds and surgical site wounds[20] and are on the Global Priority List released by the World Health Organization due to their threat to human health and their increasing resistance towards antibiotics[21–23].

In this study, we present a computational workflow leveraging the inherent clustering of peptides to simplify peptidomic data analysis. By employing a community-based algorithm, we capture the natural clustering of peptides into entities denoted as peptide clusters. The clustering of peptides enables the implementation of reliable and large-scale analytical methods akin to those employed in other omics fields. This inherent dimensionality reduction enhances inter-sample comparability by lessening the problem of missing values. Further, the workflow enables the detection of phenotype-specific peptide regions, shedding light on differing protein degradation patterns. We apply this method to deconvolute and study the peptidomes from porcine wound fluids infected by *S. aureus* and *P. aeruginosa*, to uncover the infected wound fluid peptidome and find pathogen-specific peptide clusters during the earliest stages of bacterial colonization. Additionally, we demonstrate that our method generalizes to other systems by analyzing complex clinical non-healing wounds in human patients. Lastly, we demonstrate its utility and generalizability on data from patients with type 1 diabetes, where we show that the clustering results in more informative features with fewer missing values which increases the classification accuracy. The methodology underlying the creation and analysis of peptide clusters has been provided in a Python package available under an MIT license[24].

## Results

In clinically infected wounds, the bacterial composition depends on the patient's microbiome as well as other factors, such as pre-existing conditions and patient genotype[25]. Further, the time of initial infection is often unknown, and the composition of bacterial species varies, complicating investigations of bacterial infection dynamics on the peptidome. To mitigate these challenges, we used wound fluids from porcine wounds infected with *S. aureus* ($N = 21$) and *P. aeruginosa* ($N = 17$) on day 0. Four of the *S. aureus*-infected wounds were infected with *P. aeruginosa* on day 1, resulting in a double infection. 13 control wounds were not infected. Sample identity specifications can be found in Supplementary Table 1. The wounds were covered with a dressing that absorbed the wound fluid, which was changed every 24 hours and analyzed over a time course of 2–3 days from infection (Fig. 1a)[26]. Proteins were separated from the peptidome using molecular weight

cutoff filters of 30 kDa, whereby the wound-derived peptides were identified and quantified by liquid chromatography-tandem mass spectrometry (LC-MS/MS)[27]. In total, 14950 peptides from 539 proteins were identified across all samples, of which 3337 were exclusively identified in a single sample, demonstrating the relatively low overlap between samples. On average, each sample contained $90.0 \pm 6.5\%$ missing values. Most peptides were identified in *P. aeruginosa*-infected wounds and the fewest were in the control wounds. Characterization of the wound fluid peptidomes can be found in Supplementary Notes 1 and Supplementary Fig. 1.

Closer inspection of the peptidome revealed peptide clusters comprised of peptides with partially overlapping sequences, differing by single terminal amino acids, compatible with the activity of exo-peptidases. These variants introduce redundancy in the dataset and complicate inter-sample comparisons. Although methodologies aiming to cluster linear peptide sequences have previously been developed to reduce the complexity of epitope-related data[28–30], they do not take peptide length and proximity in the protein backbone into account, which is crucial for the clustering of the peptidome. Therefore, we developed an algorithm that clusters similar peptide sequences with respect to their proximity on the protein backbone.

To initialize clustering, protein-centered peptide networks were generated for each protein separately by connecting peptides with overlapping sequences, similar lengths, and a centroid distance below a certain threshold (Fig. 1b, Eqs. 1–5 in Methods, Supplementary Notes 2 and Supplementary Fig. 2). The topology of the resulting networks depends on the peptide content as well as parameter choices in the clustering algorithm, and may result in small islands of distinct and highly separated clusters, or large connected components as a result of the continuous overlap between peptides derived from different cut sites. To separate the connected components, the resulting networks were further partitioned by applying the Leiden community detection algorithm[31], seeking to maximize the modularity of the network (Fig. 1b). This partitions the large connected components into highly connected subcomponents, which effectively separates between the unique clusters. A description of the implications of parameter choices during clustering can be found in Supplementary Fig. 2 and Supplementary Notes 2. A uniform manifold approximation and projection (UMAP) of the network for apolipoprotein A1 (APOA1) is shown as a showcase in Fig. 1c. Another visualization of the clustering when projecting the peptides in APOA1 to the protein backbone can be seen in Fig. 1d.

### Characterization of wound fluid peptidomes

To characterize the wound fluid peptidomes, we applied our community-based clustering strategy to the complete porcine wound fluid dataset of 13,259 unique peptide sequences, resulting in 743 clusters, thereby reducing dimensionality by ∼95%. The average number of missing values was reduced by $70 \pm 13\%$, representing an increase in present values by $375 \pm 100\%$. Peptides were quantified by the top 3 most intense peptides. Grouping the peptide clusters using hierarchical clustering results in three distinct groups. One group predominantly contains shared peptide clusters found in most samples whereas the other groups contain peptide clusters abundant in infected samples or a smaller set of samples (Fig. 2a). Dimensionality reduction by UMAP showed that the quantified clusters separate the sample types into distinct groups (Fig. 2b). In general, peptide clusters are more abundant and numerous in infected wounds, both in terms of intensity and number of unique clusters (Fig. 2c), indicating higher proteolytic activity during infection. The *P. aeruginosa*-infected wounds contain more abundant and unique clusters compared to those infected by *S. aureus*, while the double-infected wounds show similar cluster intensities as wounds infected by *P. aeruginosa*. *P* values were calculated using linear regression and corrected for multiple hypothesis testing with the Benjamini-Hochberg procedure. In total,

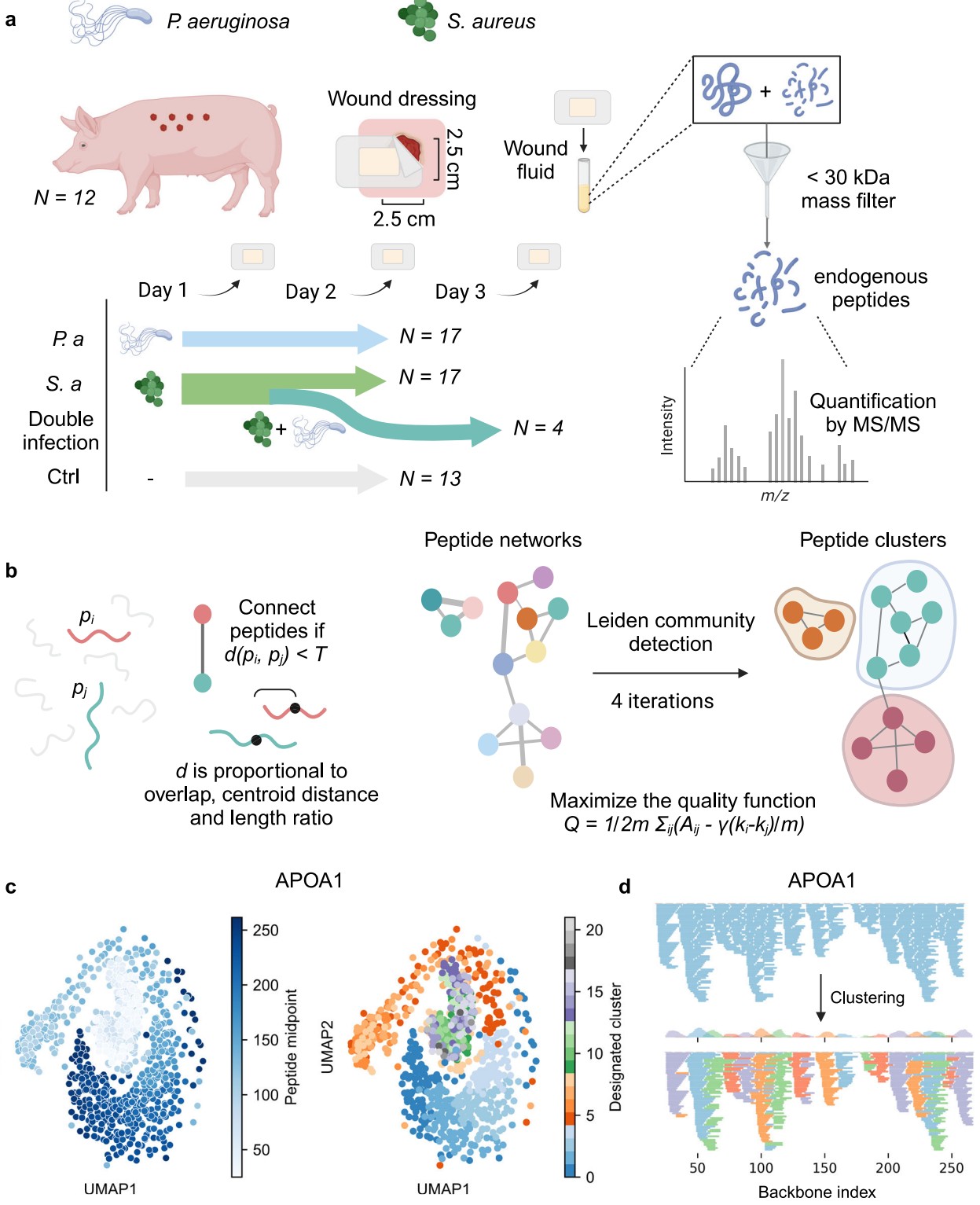

68 clusters from 33 proteins had a $|log_2(\text{fold change})| > 2$ and a $Q$ value $< 0.05$ when comparing wounds infected by *P. aeruginosa* and *S. aureus*, demonstrating that the peptide clustering strategy enables the identification of clusters that are differentially abundant between sample types. In Fig. 2d, the abundances of the 5 top-scoring clusters by $Q$ value in *S. aureus* and *P. aeruginosa* respectively are shown.

Importantly, the 20-fold decrease in the number of peptidomic features enables standard omics-based machine learning-based classification. Here, we trained an XGBoost[32] classifier to distinguish between the controls, *S. aureus*, and *P. aeruginosa*-infected samples

and achieved a classification accuracy of $94 \pm 2\%$ (Fig. 2e). To investigate how far the dataset can be reduced before losing classification accuracy, a modified version of recursive feature elimination using SHAP[33] was applied to iteratively remove features deemed least important to the classifier. Optimal performance was achieved when utilizing 10–60 clusters (Fig. 2f).

Infections are not binary but exist on a continuum, as some wounds are more heavily colonized by certain bacteria than others. We hypothesized that the amount of colonized bacteria would be reflected in the peptidome and that the peptidome therefore could be used

**Fig. 1 | Generating wound fluid peptidomic clusters. a** Wounds were created on pigs and overlaid with a dressing. The wound drainage, containing proteins and peptides, is absorbed into the dressing. The dressings were changed and sampled every day over a 2–3-day period. The fluid was extracted from the dressings, whereafter proteins were filtered out by a mass filter of 30 kDa. The endogenous peptides were subsequently identified and quantified by LC-MS/MS. In total, 21 wounds were infected with *S. aureus*, and 17 with *P. aeruginosa* on day 0. Additionally, 4 of the *S. aureus*-infected wounds were infected with *P. aeruginosa* on day 1, resulting in double-infected samples, which were sampled over days 2 and 3. In total, 13 samples were not infected and were used as a control. **b** The initial step in generating peptide clusters entails generating peptide networks (undirected graphs). Peptides are connected if they pass the threshold $d\left(P_i, P_j\right) < T$ where $d$ is a function of peptide overlap, length ratios, and centroid distance, and $T$ is a chosen threshold. Here, an optimal threshold of 4 was identified empirically and applied to generate the networks (see Supplementary Notes 2 and Supplementary Fig. 2 for more information). The resulting network is further partitioned by applying the Leiden community detection algorithm[31], which seeks to maximize the modularity of the network, $Q$, to finally create peptide clusters. **c** UMAPs of the peptide networks of APOA1. Each scatter represents a peptide and is colored by the starting position (left) and designated cluster after applying the Leiden algorithm (right). **d** Visualization of the clustering algorithm when applied to APOA1. The upper panel shows all the peptides without clustering projected onto the protein backbone. The lower panel shows the results after clustering, where peptides from each cluster are colored differently from their neighbors depending on what cluster they belong to.

to infer the level of species-specific colonization. To test this, we used the samples from single infections as training data and mapped them onto a high-dimensional space where their coordinates were determined by the quantified clusters. The samples from double-infected wounds were then mapped onto this space and weighted k-nearest neighbor (kNN) regression was used to regress a continuous value signifying the level of likeness to the samples that were infected by *S. aureus* and *P. aeruginosa* respectively. This likeness was then correlated to the ratio between $CFU_{P.aeruginosa}$ and $CFU_{S.aureus}$. Bootstrapping ($N = 10$) was performed by adding Gaussian noise to the feature values before regressing. The correlation between the CFU-ratio and output probability was $0.93 \pm 0.03$ (Pearson r) (Fig. 2g). This demonstrates that the level of bacterial colonization is reflected in the quantified peptide clusters.

## Clusters can be utilized to identify bacterial influence on protease activity

Clustering of peptides reinforces endoprotease specificity and removes the influence of unspecific exoprotease activity. We speculated that this feature could be utilized to identify the protease activity that has given rise to the different peptidomes (see Fig. 3a, Supplementary Notes 3, and Supplementary Fig. 3 for the motivations behind the algorithm).

We first analyzed the wound fluids using zymograms. In these, different gelatinase patterns were observable in the different sample types. *P. aeruginosa*-infected wounds showed specific gelatinase activity with degradation in the 75–50 kDa and 100 kDa regions (Fig. 3b). The *P. aeruginosa*-related gelatinase activity was not observed until day 3 in the double-infected samples (Fig. 3c). The protein content was investigated with SDS-PAGE, showing little to no difference between sample types (Supplementary Fig. 4).

Investigating the p4-p4′ regions surrounding peptide terminals can reveal specific proteolytic activity in wound infections, as proteases typically cleave specifically in this window of residues. Each cluster is associated with two cut sites defined by the most frequent terminal position in each cluster and sample. The amino acid distributions in the p4-p4′ windows of these cuts were weighted by the mean peptide intensity in the cluster, generating a weighted amino acid distribution for each sample at each position surrounding the cut site. The weighted amino acid distributions of infected samples were then compared to the control distributions generated from the control samples using the Kullback-Leibler (KL) divergence. Projecting the KL-weighted cut site amino acid frequencies to two dimensions using UMAP separates the samples cluster based on infection type, but not on time point, contrary to what was observed when projecting the quantified clusters (Fig. 3d).

Across all infected sample types, the largest divergence was observed in the p1-position, indicating that the altered protease activity is specific at this position. The *P. aeruginosa*-infected samples have a cut site preference for lysine at the p1-position, whereas *S. aureus*-infected wounds were specific for valine and alanine. In *S. aureus*-infections, there was also a specificity at the p1′ position, although not as distinct as the one at p1. The double-infected wounds

exhibited an *S. aureus*-like profile on day 2, whereafter the typical *P. aeruginosa* lysine-specificity emerged on day 3 (Fig. 3e) which correlates with the patterns in the zymograms. We quantified this transition by the total KL divergence as shown in Fig. 3h, which reiterates that *P. aeruginosa*-colonization is delayed when there is a pre-existing *S. aureus*-colonization.

In total, 160 peptide clusters were uniquely identified in the *P. aeruginosa*-infected wounds over the complete period. Most of these clusters were flanked by cut sites with lysine at the p1-position (Fig. 3e). A similar profile emerged on day 3 in the double-infected samples (Fig. 3f). These results show that the peptide clusters and their intensities change in a time-dependent fashion, whereas the cleavage patterns are largely similar over time and are dependent on the predominant type of bacterial pathogen in the wound.

## Pathogen-specific peptide clusters

The previous analyses have demonstrated that the algorithm enabled the identification of differentially abundant clusters and can provide evidence of bacteria-specific proteolytic activity in the peptide content of infected wound fluids. In Fig. 4, we present three proteins' differential degradation patterns depending on the type of infection. These proteins were selected due to having differentially abundant or unique peptide clusters reflecting the presence of a pathogen.

Firstly, we highlight the N-terminal region 0–16 of hemoglobin subunit alpha (HBA). This is the most differentially abundant cluster in *P. aeruginosa* when quantifying the clusters based on the top 3 most differentially abundant peptides within the respective cluster (Fig. 4a). Further, a lysine is situated at the p1-position of the cut site, consistent with the most frequent cut site in wounds infected by *P. aeruginosa* (Fig. 4b). Secondly, we highlight the cleavage of PR-39. Peptides in the antimicrobial region of PR-39 were more abundant in wounds infected by *P. aeruginosa*, while the peptides from the rest of the protein were more abundant in wounds infected by *S. aureus*. Lastly, we discovered clusters that were only present in one infection type, such as clusters derived from high mobility group protein B1 in wounds infected by *P. aeruginosa*. The cleavage pattern emerges on day 3 in the double-infected samples, consistent with the increase of *P. aeruginosa*.

## Identification of subtle proteolytic differences in contaminated wounds

Culturing of the wounds revealed that four of the wounds infected with *S. aureus* on day 0 were contaminated prior to inoculation[26] (Fig. 5a). The enzymatic activity in these wounds was analyzed with zymograms, showing similarities to patterns typical for *P. aeruginosa* on day 1, and *S. aureus* on day 2 (Fig. 5b). These results were corroborated with peptidome analysis of the wounds to demonstrate the applicability of the peptide clustering workflow. A UMAP of the quantified clusters revealed that the samples cluster with *P. aeruginosa* samples on day 1, whereafter they shift to being more *S. aureus*-like (Fig. 5c). Performing differential abundance analysis between day 1 and day 2 of the contaminated samples shows a significant shift in the amount of HBA (0–16) with a $Q$ value < 0.05 (Fig. 5d).

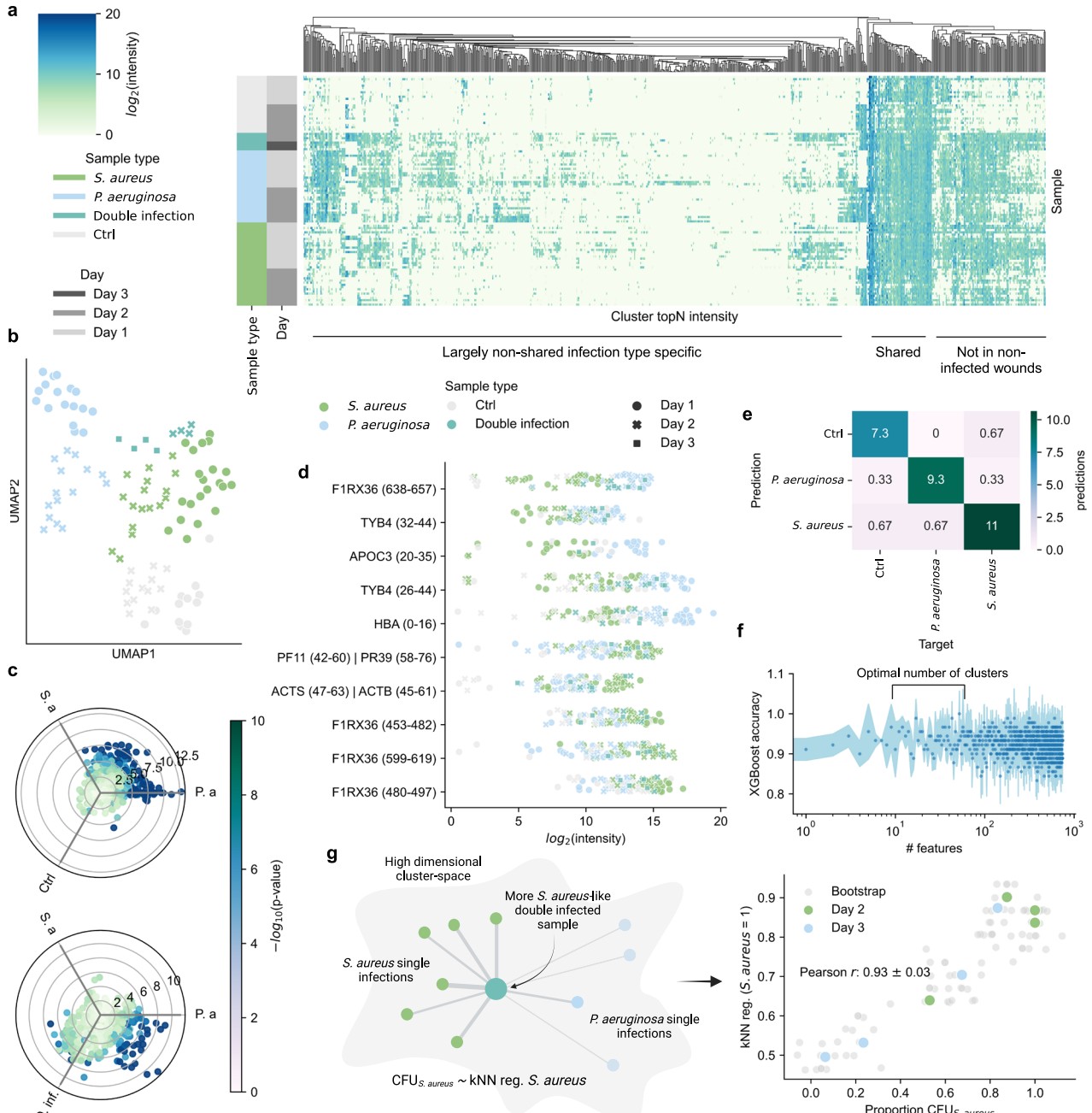

**Fig. 2 | Peptide clusters enable large-scale omics-esque analysis. a** Clustermap of the 743 peptide clusters. The color intensity is proportional to the cluster intensity calculated by the topN method on the $log_2$-transformed intensity. Hierarchical clustering was performed on the peptide clusters (columns). **b** UMAP of the quantified clusters colored by sample type. The shape indicates the day the samples were taken. **c** A polar projection where each scatter represents a cluster. The position of the cluster is computed as $\sum \hat{v}_i$ where $i$ are the sample types $|\hat{v}_i|$ the cluster quantity. Each sample type is separated by 120°. The color intensity is proportional to the $-log_{10}(p - value)$ as determined by one-way ANOVA, clipped to $\leq 10$. **d** Stripplot of the $log_2$-intensities of the top 5 clusters by $Q$ valu**e** for *P. aeruginosa* (upper) and *S. aureus* (lower). **e** Average confusion matrix from k-fold cross-validation ($k = 3$) of an XGBoost classifier. **f** Recursive feature elimination using SHAP to estimate feature importance was performed to investigate the number of optimal features for an XGBoost classifier. The mean accuracy after

removing a given number of features is shown as scatters. Error bands represent $\pm 1$ SD. An optimal number of features was determined to be in the range of 10–60 features (out of 743 total). **g** The peptidomic similarity of double-infected wound fluid peptidomes to single-infected peptidomes was evaluated using weighted kNN regression. The illustration shows how a double-infected sample lies closer to *S. aureus*-infected samples in the high-dimensional cluster space and is, therefore, more *S. aureus*-like. The *y* axis in the scatter plot showed the likeness towards *S. aureus* samples when the double-infected samples were mapped onto the cluster space. The *x* axis shows the proportion of CFU for *S. aureus*. Bootstrapping was conducted 10 times by adding noise to the training data followed by regression. The correlation between the bootstrapped regressed likeness and *P. aeruginosa* CFU-proportion is 0.93 ± 0.03 (Pearson *r*). Source data are provided as a Source Data file.

Notably, the cut site specificity on day 1 resembled that of *P. aeruginosa* or the double infection on day 3, with lysine being the most common amino acid at p1 and the divergence against *S. aureus* is high. The cut site patterns change on day 2, where they resemble *S. aureus*

more (Fig. 5e). Furthermore, using the same kNN regressor that was used to estimate the CFU-composition above, demonstrates a similar pattern to *P. aeruginosa* on day 1, which then shifted to *S. aureus* on day 2 (Fig. 5f). The cut site specificity was quantified by computing the total

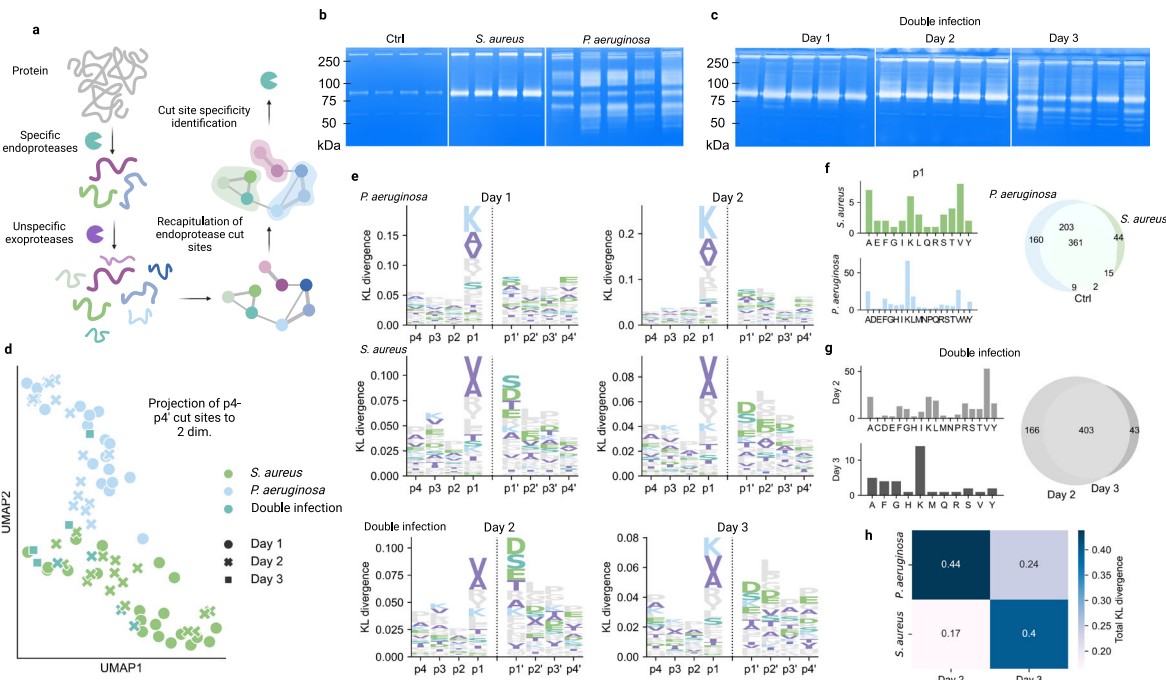

**Fig. 3 | Cut-site analysis reveal pathogen-specific protease patterns. a** Graphical representation illustrating how peptide clusters facilitate the identification of endoprotease specificity by recapitulating the endoprotease cut-sites and increasing the signal-to-noise ratio. **b, c** Zymograms for representative single infections and control samples for day 1 (**b**) and for all double-infected samples (**c**). Uncropped zymograms for all infected wounds and time points are available in the source data. Double-infected samples on day 1 were just infected by *S. aureus*. An approximate kDa ladder is shown to the left of the gels. **d** Projection of KL-weighted p4-p4′ amino-acid frequency for all samples onto two dimensions using UMAP. **e** Logoplots of cut site specificity for the p4-p4′ window. The height of the column is calculated by the KL divergence between the weighted amino acid distributions for the infected sample types against the control. The amino acid distributions were generated by identifying the amino acid at the most commonly occurring termini for each peptide cluster. **f** The left panels show the p1 amino-acid distributions of the cut sites for the clusters found uniquely in the *S. aureus*-infected wounds and in *P. aeruginosa*-infected wounds, respectively. The right panel shows a Venn diagram of the overlap of peptide cluster identifications. **g** Same layout as in **f**, but for the double-infected samples. **h** The sum of the total KL divergence in the p4-p4′ window for the double-infected samples against *P. aeruginosa* and *S. aureus* across days 2 and 3. Source data are provided as a Source Data file.

KL divergence in the p4-p4′ window, showing that the divergence to *S. aureus* decreases from day 1 to day 2, while there was an increase towards *P. aeruginosa* (Fig. 5g). Together, this demonstrates that the methodology can be applied to identify differences in the proteolytic patterns resulting from bacterial contamination, which mimics the less controlled conditions of real-world data.

### Identification of proteolytic differences in human non-healing wounds

Lastly, we investigated whether the methodology developed here generalizes to human samples by applying it to human wound fluids from patients with non-healing leg ulcers. Common bacteria found in such non-healing wounds typically include *S. aureus*, *P. aeruginosa,* and *Enterococcus* species[34]. A total of 18 samples were analyzed with the LC-MS/MS workflow described above, resulting in 40,845 peptides. Ten of the samples were primarily colonized by *P. aeruginosa*, and 8 by *S. aureus*. The qualitative bacterial composition of the wounds was determined using MALDI-TOF, and the protease activity was analyzed with zymograms (Fig. 6a, b). The complete list of identified bacterial species and sample specifications can be seen in Supplementary Table 2 and described in Supplementary Notes 6.

Peptide clusters were generated with the same settings as previously applied to the porcine samples, finally resulting in 3020 clusters, thereby reducing the dataset size by 93% and increasing the fraction of present values by $262 \pm 43\%$. Cut site analysis revealed a highly specific motif at the p1-position. The motif is identical to that of human neutrophil elastase (NE), with a strong specificity for valine, alanine, isoleucine and threonine, in descending order[35,36] (Fig. 6f). Dimensionality reduction using UMAP was performed on the KL-weighted p4-p4′ amino acid frequencies, revealing three clusters with different degrees of NE-like p1 specificity (Fig. 6g). All *P. aeruginosa*-infected wounds revealed a moderate or high level of such specificity, while most *S. aureus*-infected samples had a low level of NE-like specificity (Fig. 6d).

Performing hierarchical clustering on the quantified peptide cluster intensities revealed that they largely group by NE-like activity and bacterial species (Fig. 6c, Supplementary Fig. 5). However, samples 9, 13, 15, and 16 cluster with the *S. aureus*-infected samples even though *P. aeruginosa* was identified by MALDI-TOF. In this context, it should be noted that MALDI-TOF mainly provides bacterial identification and not quantification, and hence, lower amounts of *P. aeruginosa* may have been present in these wounds. Interestingly, three of these samples have either been treated with antibiotics for the ongoing infection or show no sign of *P. aeruginosa* by zymogram appearance, forming a new subphenotype. Differential testing between the *P. aeruginosa*-infected samples and the new subphenotype, against the *S. aureus*-infected samples highlight clusters that are unique to this subphenotype (Fig. 6e). While the dataset presented here is limited, these showcase the capability to identify clinically relevant subphenotypes using a peptidomics approach.

The clusters exclusively identified in the samples with high NE-like specificity were flanked by cut sites with valine, alanine, and isoleucine at the p1-position. Similarly, the clusters exclusively present in the moderate NE-samples which are mainly colonized by *P. aeruginosa* showed a specificity towards lysine in the p1-position (Fig. 6g), which was also identified when stratifying on primary bacterial colonizer (Supplementary Fig. 5b). In conclusion, these results show that the peptide cluster strategy proposed here generalize to complex non-healing human wounds and can enable identification of phenotype-

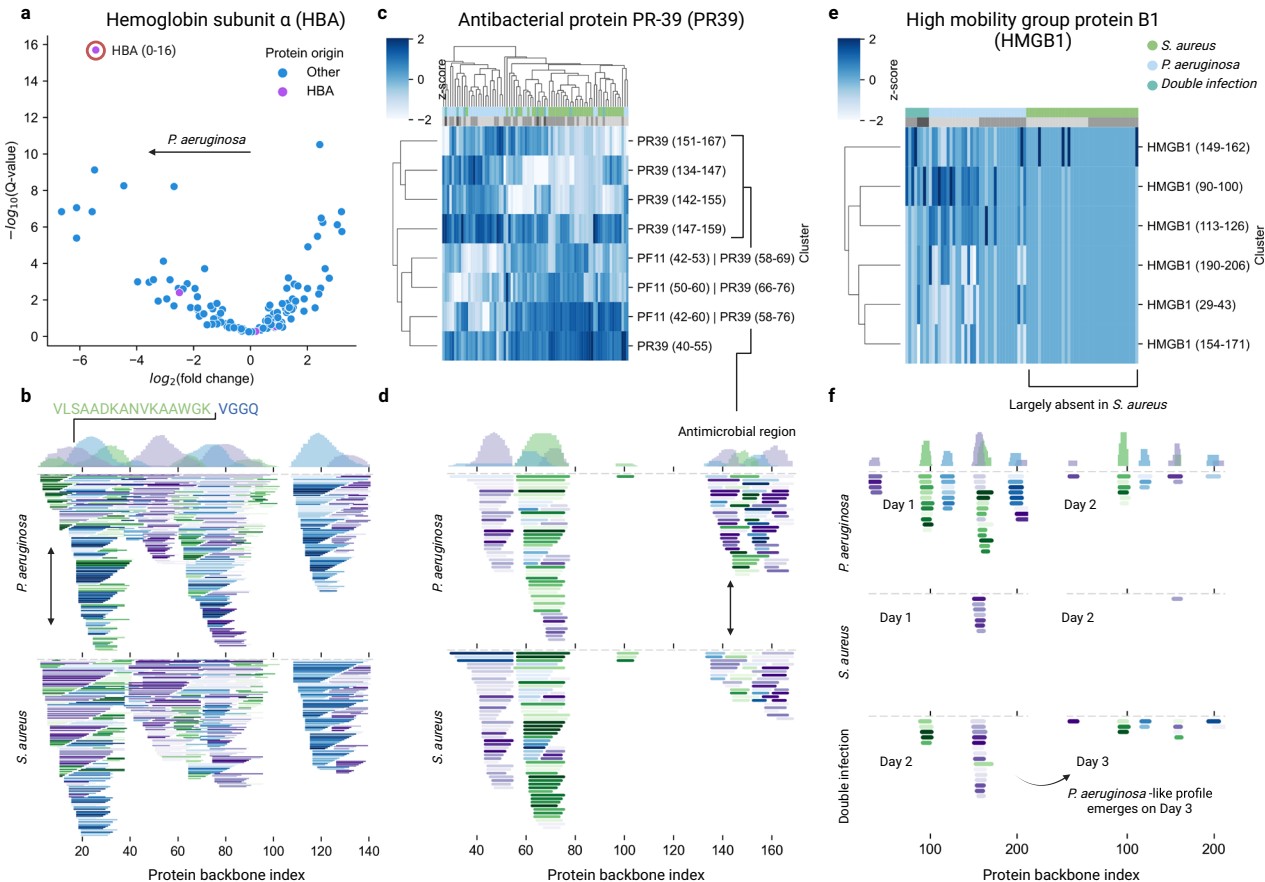

**Fig. 4 | Identification of potential pathogen-specific peptide clusters. a** Volcano plot where HBA (0–16) is highlighted after differential abundance analysis between samples from wounds infected by *P. aeruginosa* and *S. aureus*. Clusters were quantified by the top three peptides exhibiting the largest differential expression score. **b, d, f** shows a visualization of the peptidome projected onto proteins. In all these projections, the top panel contains a histogram showcasing the number of identified residues at a given position, separated on peptide clusters. The clusters are colored repeatedly in green, blue, and purple. The color intensity is proportional to the mean peptide intensity. **b, d** The samples for single infection over days 1 and 2 are pooled. **b** Visualization of the peptidome projected onto HBA, where the initial cluster corresponds to HBA 0–16. **c** Clustermap of the clusters from PR-39. Some clusters are non-proteotypic and also map to platelet factor XI (PF11). **d** The PR-39 peptidome mapped onto the protein backbone, where the antimicrobial region is highlighted. **e** Clustermap of the clusters from high mobility group protein B1 (HMGB1), similar to **c**. **f** HMGB1 peptidome projected onto the backbone of HMGB1. We highlight the re-emergence of the *P. aeruginosa*-like degradation pattern on day 3 in the double-infected samples.

specific clusters as well as the characterization of the proteolytic environment.

## Comparison with traditional peptide-centric analyses

To evaluate the efficacy and generalization of our computational methodology, we applied it to a publicly available dataset from a study by Van et al.[37], downloaded from ProteomeXchange. This dataset comprises 30 urine samples analyzed using MS/MS peptidomic methodology, including 15 samples from patients with type 1 diabetes and 15 control samples. We re-created part of the analysis (Fig. 2e, f in ref. 37), as shown in Fig. 7a–c. *P* values were calculated using linear regression and corrected with the Benjamini–Hochberg method for multiple hypothesis testing.

We then compared these results with our methodology (Fig. 7d). Clustering was performed with a cutoff of 4 and a resolution of 0.8, consistent with our other analyses. Clusters were quantified using the top 3 peptides with the highest differential expression (DE) scores. Our findings revealed that the uromodulin (UMOD) peptides identified by the classical approach are from the same cluster (UMOD 458–474). Additionally, we identified two new clusters (LMAN2 161–177 and SECTM1 118–139) that passed the *Q* value cutoff of 0.05 (Fig. 7e). Interestingly, none of the peptides from these proteins were included in the top 15 peptides identified with a peptide-centric approach. Out

of 43 clusters passing the *p* value cutoff of 0.05, the top 20 clusters are visualized in Fig. 7f for clarity. The clustering approach revealed two apparent subgroups of type 1 diabetes that were not apparent using the traditional peptide-centric approach. Further analysis using UMAP projected the subsetted feature matrices to two dimensions, showing clear separation by sample type in both peptide-centric and cluster-centric workflows (Fig. 7g).

To illustrate the improvement in feature quality through clustering, we performed an iterative scheme to evaluate the number of missing values and the machine learning-based predictive power of progressively larger feature matrices. The original feature matrices were subsetted to include the top N features sorted by *p* value, beginning with $N = 3$. At each iteration, a feature was added. The number of missing values was plotted against the normalized number of features included in the feature matrices in Fig. 7h. The clustered data contains a smaller proportion of missing values among the most discriminating features compared to the peptide-centric data since these clusters utilize quantification information from a combined set of peptides. Secondly, we evaluated the area under the receiver operating characteristic curve (AUROC) across progressively larger feature matrices. 10 logistic regression classifiers were trained on different stratified subsets of half the dataset, and the AUROC was evaluated on the remaining halves. This iterative process continued until the full dataset size was reached. The

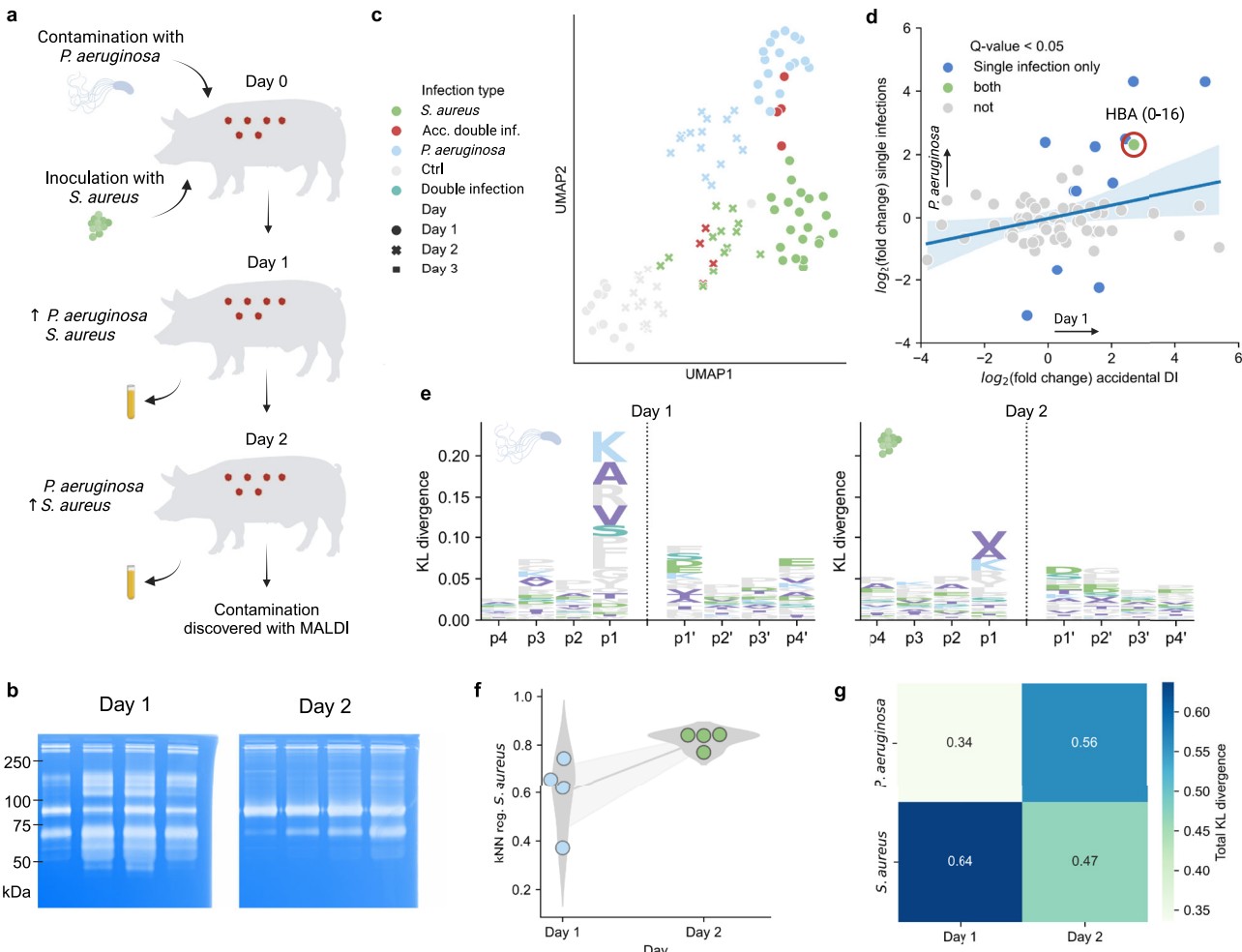

**Fig. 5 | Peptidomic investigation of contaminated wounds. a** Illustration depicting the origins of the contaminated samples. **b** Zymograms of the contaminated wounds across days 1 and 2. An approximate kDa ladder is shown to the left. Uncropped gels can be found in the source data. **c** UMAP of quantified clusters, with samples colored by type and shaped according to the day of sample collection. **d** Comparison of the differential abundant clusters between the single infections and the days of the contaminated samples. The scatters are colored based on which combination of comparisons they exhibit a *Q* value < 0.05. Linear regression was performed and is showcased with a 95% confidence interval estimated using a bootstrap. **e** Logoplots illustrating cut site specificity (p4–p4′) for contaminated

samples. Column height is determined by the KL divergence between weighted amino acid distributions for samples against *S. aureus* distributions. **f** The likeness of the contaminated samples to *S. aureus* as determined by kNN regression. The line shows a linear regression for the likeness across day 1 and 2, alongside a 95% confidence interval estimated using a bootstrap. **g** Heatmap showing total KL divergence between the amino acid distributions of contaminated wounds (with days as columns) and single infections (pooled over days 1 and 2). The total divergence against *P. aeruginosa* increases between days 1 and 2, while divergence towards *S. aureus* decreases. Source data are provided as a Source Data file.

resulting AUROCs across the feature space are shown in Fig. 7i. The AUROC of the clustered data was consistently higher than the peptide-centric data. This demonstrates that clustering combines peptides with missing information into more informative features.

A comparison to a traditional workflow on the porcine dataset is presented in Supplementary Notes 3 and Supplementary Fig. 3.

### Replicability and robustness of data
Due to the high number of missing values in the unprocessed dataset, we assessed the replicability and robustness of our data. We conducted a blinded re-run of a subset of samples, which confirmed the technical reproducibility of our findings (see Supplementary Notes 4 and Supplementary Fig. 6). Additionally, we applied down-sampled bootstrapping to the peptide cluster feature matrix before performing dimensionality reduction with UMAP. This analysis showed that missing values had little impact at the peptide cluster level, as the UMAP projections were still clustered by sample type and time point (Supplementary Notes 5 and Supplementary Fig. 7).

### Discussion
Understanding the impact and the underlying processes behind protein degradation has begun to enhance our understanding of biological systems but represents a largely unexplored resource for the identification of new biomarkers and therapeutic targets. However, large-scale peptidomics analyses present challenges rooted in the inherent complexity and scale of the peptidomic landscape. To address these challenges, we developed a method that deconvolves the peptidome by using networks and community detection algorithms to approximate optimal partitioning. The algorithm clusters overlapping peptides and reduces the complexity of the peptidome datasets by 93–95% which opens up new avenues of analytical strategies similar to those successfully employed in other omics fields. The method also enables a new definition of an endoprotease cut site as the most common terminal to a peptide cluster, leading to improved cut site analyses by increasing the signal-to-noise ratio. Importantly, we demonstrate that classification models can utilize peptide clusters to distinguish between samples, identify important clusters, and estimate

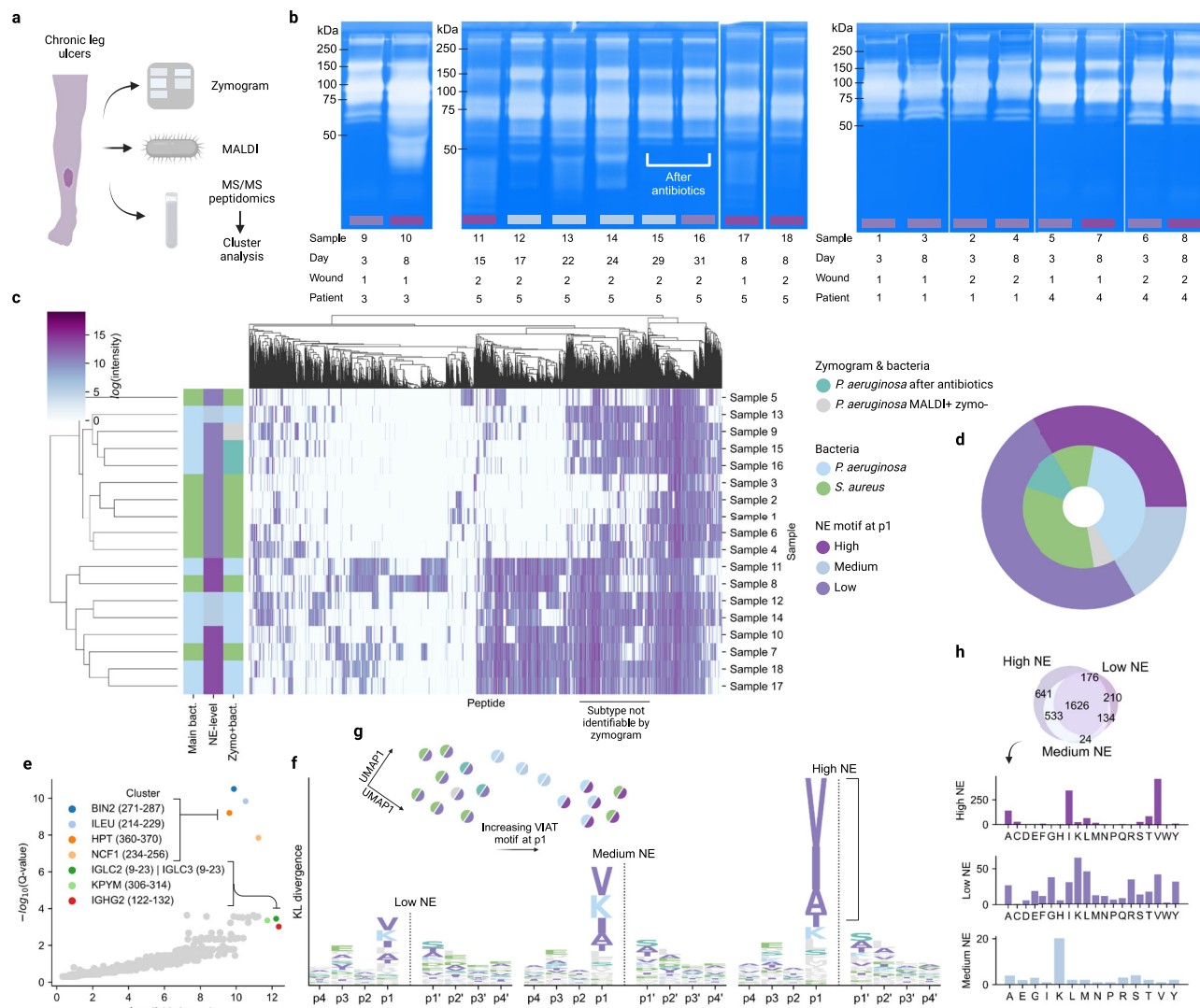

**Fig. 6 | Characterization of human chronic leg ulcer wound fluid peptidomes.**
**a** Schematic illustration sample preparation workflow. **b** Zymograms of the 18 samples. The colored bar beneath each lane indicates the level of NE-like cut sites. An approximate kDa ladder is depicted beside the gels. Uncropped gels are available in the source data. **c** Clustermap displaying log-transformed cluster intensities quantified by the top three most intense peptides. Row colors correspond to the primary bacterial colonizer identified by MALDI-TOF, NE-level, and phenotype determined by zymogram/MALDI-TOF. **d** Pie chart of bacterial phenotype determined by zymogram/MALDI-TOF (inner) and NE-like specificity levels

(outer). **e** Volcano plot resulting from differential abundance analysis comparing samples 1–6 versus 7–18. Clusters with the lowest $Q$ value and highest fold change are highlighted. **f** Logoplots illustrating the mean p4-p4' amino-acid frequency for three sample groups identified by clustering, corresponding to distinct levels of NE-like specificity at the p1-position. **g** UMAP representation of the p4-p4' amino-acid frequency for all samples, with coloring indicating NE-like specificity (right half) and zymogram/MALDI phenotype (left half). **h** Amino-acid distribution identified at the p1-position for unique clusters when comparing samples with differing levels of NE-like specificity. Source data are provided as a Source Data file.

the type and amount of pathogen-specific colonization and activity. The method advances our understanding of the peptidome by enhancing the interpretability of large-scale peptidomics data in complex biological contexts. Further, it generalizes to other datasets such as that of urine peptidomics from type 1 diabetics[37].

The data and algorithm presented here have several shortcomings that warrant consideration. Firstly, the algorithm is dependent on the manual selection of parameters, such as the threshold used for creating the networks and the resolution parameter, which is important for partitioning. The values for these parameters were evaluated empirically by visual assessment (see a description of the process in Supplementary Notes 2 and Supplementary Fig. 2). The reason was related to the challenges associated with finding a data-driven parameter selection algorithm to generate optimal clustering, mainly because the optimization goal was difficult to define. Secondly, the datasets presented here are limited, with 111 porcine samples and 18

human samples. Although the datasets are comparably large compared to other previously published datasets, the complexity and diversity of wounds and their peptidomes call for further studies to validate the findings.

Despite these limitations, we identified cut sites that were specific for the primary wound colonizers. For example, wounds infected by *P. aeruginosa* were highly specific for lysine residues at p1 which fits with the reported specificity of the secreted virulence factor protease IV[38]. In the human non-healing wounds, we discovered cut-sites matching the human NE-like proteases which is in agreement with previous studies[39]. Although we did not identify any known proteases that fit the cut site specificity of *S. aureus*, we identified several differentially abundant and unique clusters in both cases. A prominent example is the N-terminal cluster 0–16 from HBA which was significantly differentially abundant in *P. aeruginosa*-infected porcine wounds.

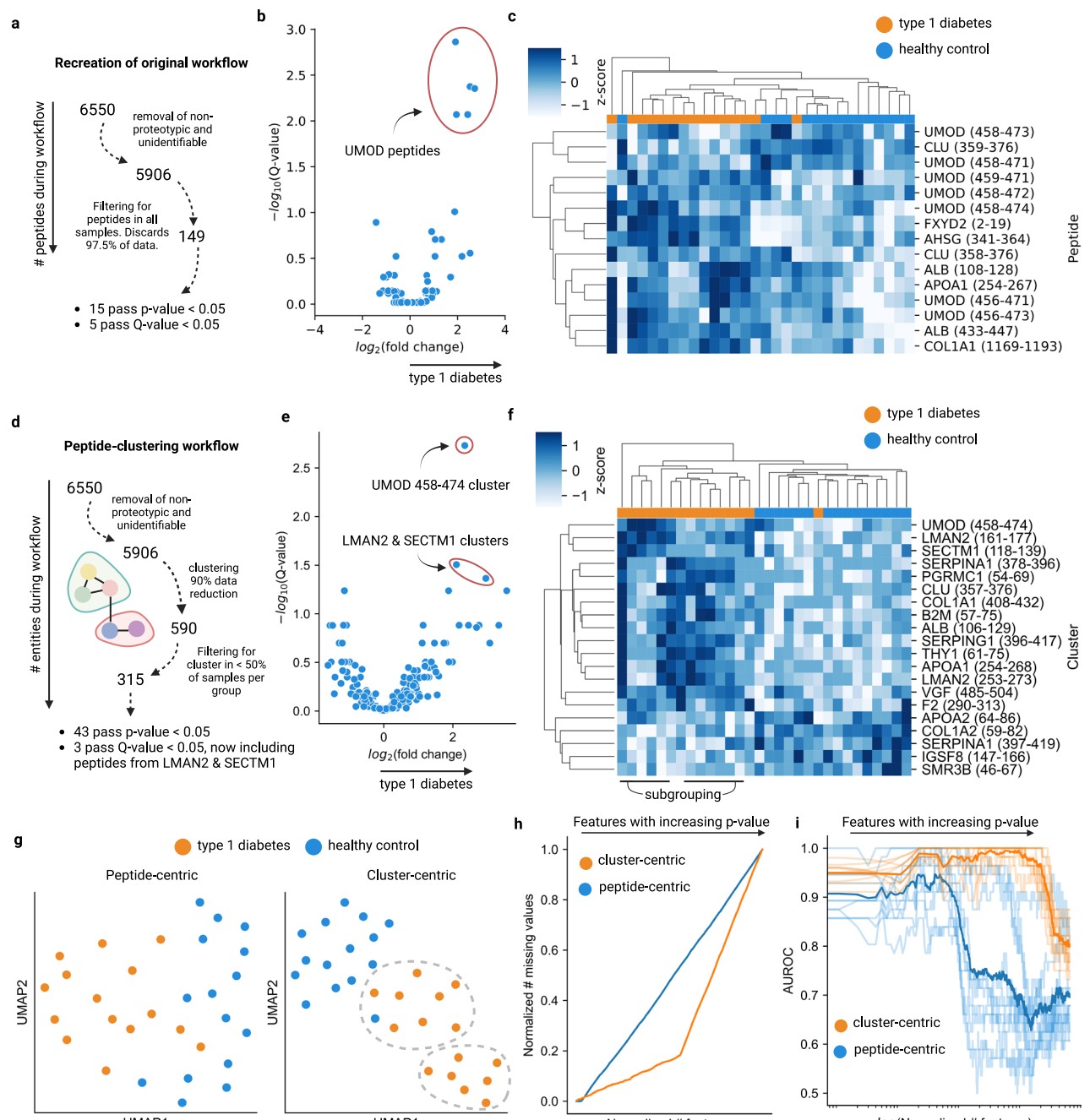

**Fig. 7 | Comparing a peptide-centric workflow with a cluster-centric workflow on a publicly available dataset. a** Workflow re-creation from Van et al., filtering peptides present in all samples to yield a final dataset of 149 peptides. **b** Volcano plot corresponding to Fig. 2e in Van et al., displaying $Q$ values on the $y$ axis and highlighting 5 UMOD peptides. **c** Clustermap replicating Fig. 2f in Van et al. **d** Outline of the peptide cluster workflow. Clusters were included if present in at least half the dataset, resulting in 315 clusters quantified by the top 3 most differentially abundant peptides. **e** Volcano plot of quantified clusters, annotating clusters with $Q$ value < 0.05. **f** Clustermap of the top 20 clusters by $Q$ value, highlighting two subgroups identified by hierarchical clustering. **g** UMAP projections of subsetted peptide and cluster matrices from **c** and **f**, highlighting subgroups identified in the cluster-centric UMAP. **h** Normalized number of missing values plotted against the normalized number of features in iteratively larger feature matrices, where features with larger $p$ values are iteratively added. $P$ values were calculated using linear regression. **i** AUROC versus the number of included features, comparing logistic regression classifiers trained on iteratively larger feature matrices for both peptide-centric and cluster-centric workflows. 10 classifiers were trained for each feature addition. All results are shown as faded lines, and the non-faded lines represent the means. Source data are provided as a Source Data file.

In conclusion, the methodology presented here has the potential to instantiate a new branch of peptidomics research by enabling large-scale data-driven analyses to identify peptidomic signatures in different biological contexts. We envision similar methodologies could be applied to a plethora of contexts such as infectious diseases, cancer, and metabolic disorders.

## Methods

### Ethics statement

All animal experiments are performed according to Swedish Animal Welfare Act SFS 1988:534 and were approved by the Animal Ethics Committee of Malmö/Lund, Sweden (permit number M131–16). The use of human wound materials was approved by the Swedish ethical

review authority (etikprövningsmyndigheten application number 2023-05051-02). Written consent was received from all subjects prior to participation.

## Porcine wound samples

In a previous study exploring the effects of the thrombin-derived antimicrobial peptide TCP-25, partial thickness wounds of Göttingen minipigs were either infected with *S. aureus* or *P. aeruginosa*[26]. The wounds were then dressed with a polyurethane dressing, which was changed 24 and 48 hours post infection, and the old dressings were collected for wound fluid extraction. The same dressing procedure was also performed for uninfected control wounds as well as double-infected wounds, which were first infected with *S. aureus*, and infected with *P. aeruginosa* after 24 hours, during the first change of polyurethane dressing. The dressing of the double-infected wound was also collected 72 hours post *S. aureus* infection[26]. The collected dressings were placed in syringes and soaked in 10 mM Tris (pH 7.4), before ejecting the fluids. Halt Protease Inhibitor Cocktail (Thermo Fisher Scientific, USA) was then added to half of the extracted wound fluid from each sample. All samples were then stored at −80 °C before use.

## Quantitative bacterial counts

The swabs and dressing fluid samples were diluted with sterile PBS to generate 7 10-fold serial dilutions from $10\times$ to $10^{7}\times$. Six separate 10 µL drops of the undiluted sample and each of the dilutions were deposited on a Todd-Hewitt agar plate. The plates were incubated at 37 °C in 5% CO2 overnight. The next morning, the number of colonies was counted and recorded.

## Wound fluids from non-healing human leg ulcers

Wound fluid from patients with venous non-healing wounds was collected from Mepilex dressings applied on the wounds for 48–72 hours (trial registration number NCT05378997). The dressings were extracted as described above, and Halt Protease Inhibitor Cocktail added as above before storage at −80 °C.

## Identification of bacteria by MALDI

Colonies from wound swab samples were prepared using the extended direct transfer sample preparation procedure on stainless steel MALDI target plates as described by the manufacturer (Bruker Daltronik GmbH). A Microflex LT/SH SMART MALDI-TOF mass spectrometry (MS) instrument with flexControl v. 3.4 (Bruker Daltronik GmbH) was used to analyze the target plate and collect mass spectra in linear mode over a mass range of 2 to 20 kDa. A spectrum of 240 summed laser shots was acquired for each sample spot. The spectra were analyzed using a MALDI Biotyper (MBT) Compass v. 4.1 with the MBT Compass Library Revision L (DB-9607, 2020) (Bruker Daltronik GmbH).

## Sample preparation, mass spectrometry, and data processing

Wound fluid extracts with supplemented protease inhibitor had their protein concentrations measured using the Pierce BCA Protein Assay Kit (Thermo Fisher Scientific, USA) according to the provided instructions. A volume corresponding to 500 µg of protein for pig wound fluids and 100 µg of protein for human wound fluid was diluted in 10 mM Tris (pH 7.4) to 100 µl, and then further diluted in 300 µl 8 M urea (in 10 mM Tris (pH 7.4) to a final concentration of 6 M urea) supplemented with 0.067% RapiGest SF (Waters, USA) (to a final concentration of 0.05% RapiGest SF). The samples were then incubated for 30 minutes at room temperature. Meanwhile, Microcon−30 centrifugal filter units were rinsed with 100 µl 6 M urea (in 10 mM Tris (pH 7.4)) by centrifugation for 15 minutes at $10,000 \times g$ at room temperature (RT). The samples were then loaded onto the filter units and centrifuged for 30 minutes at $10,000 \times g$ at RT, followed by a final rinse with an additional 100 µl 6 M urea (in 10 mM Tris, pH 7.4) and

5 minutes of centrifugation at $10,000 \times g$ at RT. The filtrate was then stored at −20 °C until LC-MS/MS analysis.

In total, wound fluids from 47 pig wounds (17 wounds infected with *S. aureus* from 4 pigs, 17 wounds infected with *P. aeruginosa* from 4 pigs, 13 uninfected wounds from 4 pigs) at two time points (24 and 48 hours post infection) had their peptides extracted. In addition, wound fluids from 4 double-infected wounds from 1 pig at three different time points (24, 48 and 72 hours post *S. aureus* infection) also had their peptides extracted. 18 human wound fluids from 4 subjects had their peptides extracted.

Extracted peptide samples were acidified by adding 1 µl 100% formic acid (FA) to 60 µl of peptide filtrate. Meanwhile, UltraMicro Spin Columns (The Nest Group, USA) were wet by adding 100 µl 100% acetonitrile (ACN) + 0.1% FA and centrifuging the column at 800 g for 1 minute at room temperature. These conditions were used for the remainder of the centrifugation steps of the solid phase extraction. The columns were then equilibrated by centrifuging 100 µl 2% ACN + 0.1% trifluoroacetic acid (TFA) through them twice. Samples were then spun onto the columns, followed by a washing step where 100 µl 2% ACN + 0.1% TFA was centrifuged through. The samples were then eluted by centrifuging 100 µl 70% ACN + 0.1% TFA through the columns. Once eluted, the samples were dried using an Eppendorf Concentrator plus at 45 °C and redissolved in 30 µl 2% ACN + 0.1% TFA.

The redissolved peptide samples were loaded onto Evotip Pure columns according to the provided instructions, apart from that the loaded samples were dissolved in 30 µl 2% ACN + 0.1 % FA instead of 20 µl 0.1% FA. These were then analyzed by LC-MS/MS on an Evosep One LC (Evosep, Denmark) coupled to a timsTOF Pro mass spectrometer (Bruker, USA). The LC was equipped with an EV1137 Performance Column − 15 cm × 150 µm, filled with 1.5 µm ReproSil-Pur C18 beads (Evosep, Denmark), and separation was performed using the accompanying 30 samples per day program. The MS used the DDA PASEF mode, doing 10 PASEF scans every acquisition cycle. The accumulation and ramp times were both set to 100 ms. Precursors with a +1 charge were ignored and the target intensity was set to 20,000, with dynamic exclusion active, at 0.4 min. The isolation width was set to 2 at 700 Th and 3 at 800 Th.

The data from the LC-MS/MS runs were searched with PEAKS X. UniProtKB reviewed (Swiss-Prot) protein list of pig proteins was used as a database when searching the pig samples, with the exchange of fibrinogen alpha chain (FIBA_PIG) and fibrinogen beta chain (FIBB_PIG) to the UniProtKB unreviewed (TrEMBL) versions F1RX36_PIG and F1RX37_PIG respectively (downloaded May 11th, 2023). When searching the human samples, UniprotKB reviewed (Swiss-Prot) protein list of human proteins was used as a database (downloaded September 29th, 2023). Data refinement was set to merge scans, correct precursor based on mass and charge states with charges between 1 and 4. It was also set to associate features with chimera scans and filter features between charges 2 and 8. During the search, the precursor tolerance was set to 20.0 ppm using monoisotopic mass and the fragment tolerance was set to 0.03 Da. Oxidation (M, +15.99) was treated as a possible modification, and a maximum of 1 modification per peptide was allowed. The search results were filtered at 1% FDR and at least 1 unique peptide for each protein. FDR was set to be estimated with decoy-function.

The peptide intensities were $\log_2$-transformed. Identified but unquantifiable peptides were imputed by sampling from a uniform distribution U(2,8) which is lower than the least abundant quantifiable peptides. The intensities were then mean-normalized so that all samples had equal intensity-means.

## Zymograms

Zymogram gels were created with a separation gel consisting of 375 mM Tris buffer (pH 8.8), 0.1% (w/v) SDS, 0.1% (w/v) gelatine, 10% (w/v) acryl amide, 0.05% (w/v) TEMED and 0.05% (w/v) APS in Milli-Q

water and a stacking gel consisting of 125 mM Tris (pH 6.8), 0.1% (w/v) SDS, 4% (w/v) acryl amide, 0.1% (w/v) TEMED and 0.05% (w/v) APS in Milli-Q water. For each sample, 5 µg of protein for porcine or 2 µg of wound fluid extract for human samples without added protease inhibitor was diluted to 5 µl with Milli-Q water, and then mixed with 5 µl sample buffer consisting of 400 mM Tris-HCl (pH 6.8), 20% (v/v) glycerol, 5% (w/v) SDS and 0.03% (w/v) bromophenol blue in Milli-Q water, which was then added to the wells. The gels were then run using an electrophoresis buffer consisting of 25 mM Tris, 200 mM glycine and 0.1% (w/v) SDS in Milli-Q water at pH 8.7 for 60 minutes at 150 V. Afterwards, the gels were washed with deionized water and incubated for 60 minutes in 2.5 % Triton X-100 at room temperature, with 160 rpm shaking, and followed by another deionized water wash. Next, the gels were incubated overnight at 37 °C in an enzyme buffer consisting of 50 mM Tris-HCl (pH 7.5), 200 mM NaCl, 5 mM CaCl2 and 1 µM ZnCl2 with 50 rpm shaking. The next day, the gels were washed in deionized water and incubated in a staining buffer consisting of 0.25% (w/v) Coomassie brilliant Blue G-250, 38.4% (v/v) ethanol and 7% (v/v) acetic acid in Milli-Q water for 60 minutes. The gels were then placed in a de-staining solution consisting of 9.6% (v/v) ethanol and 7% (v/v) acetic acid in Milli-Q and imaged using a Chemidoc MP Imaging System (Bio-Rad Laboratories, USA).

### SDS-PAGE

From each wound fluid sample, 20 µg of protein with added protease inhibitor was mixed with Milli-Q water to 8 µl. 10 µl Tricine SDS Sample Buffer (2×) and 2 µl NuPAGE Reducing Agent (10×) was added to each sample. The samples were then incubated at 95 °C for 5 minutes. 10–20% Tricine gels and running buffer were prepared as described by the manufacturer's instructions and were then run for 90 minutes at 100 V. Once the runs were finished, the gels were stained with Gelcode Blue Safe Protein Stain (Thermo Fisher Scientific, USA) according to the instructions provided by the manufacturer. Imaging was then performed using a Chemidoc MP Imaging System (Bio-Rad Laboratories, USA).

### Generating peptide clusters

The goal of generating peptide clusters is to group highly similar and proximate peptides into a single entity. Here, developed a fuzzy algorithm that utilizes networks and community detection to yield the most optimal grouping of peptides. This coincides with capturing the endopeptidase activity while filtering out the effect of exoproteases which generate clusters of highly similar peptides. The first step in generating peptide clusters entails generating weighted peptide networks, where peptides we consider to be part of the same cluster are strongly connected. In the second part, the peptide networks are partitioned by identifying communities within the network. These two steps work together to generate the optimal peptide clusters.

The distance between each pair of peptides is calculated using a distance function, $d$. The distance function devised and applied here takes peptide overlap, centroid distance, and peptide length ratios into account. The rationale behind the choice of these variables is the following:

- Peptides with a high degree of overlap are likely to belong to the same cluster.
- Long peptides with high overlap, but a large distance between their centroids are unlikely to be from the same cluster.
- Peptides with different lengths could belong to the same cluster, but should not be connected directly. If intermediate products are present, they will have an indirect connection, however, if not, they should not be connected, as they are most likely from different clusters.

The different parts of the distance function for peptides $i,j$ with starting positions $s_{i,j}$, end positions $e_{i,j}$ lengths $l_{i,j}$ and centroid positions $c_{i,j}$ are:

$$d_{overlap}\left(s_j, e_i, l_i, l_j\right) = 1/\frac{e_i - s_j}{l_i + l_j - e_i + s_j} + \epsilon \quad (1)$$

$$d_{length}\left(l_i, l_j\right) = \max\left(\frac{l_i}{l_j}, \frac{l_j}{l_i}\right) \quad (2)$$

$$d_{centroid}\left(c_i, c_j\right) = |c_i - c_j| \quad (3)$$

Where $\epsilon$ is a noise variable of $10^{-8}$. The complete distance is the sum of the factors:

$$D = \lambda_1 d_{overlap} + \lambda_2 d_{length} + \lambda_3 d_{centroid} \quad (4)$$

where $\lambda_{1-3}$ are coefficients that allow for the tuning of the weighting of the individual terms. In our study, $\lambda_1 = \lambda_2 = \lambda_3 = 1$. The peptides are connected if:

$$D < T \quad (5)$$

where $T$ is an arbitrary threshold that is purpose specific. In our data, an optimal threshold of 4 was identified empirically. Motivation for the choice of parameters and the implications of these on the clustering can be seen in Supplementary Notes 2 and Supplementary Fig. 2. A network is created for each protein separately. The created networks are partitioned into isolated components, with weaker connections between clusters with lowly overlapping components. To further partition the network, the Leiden community detection algorithm is applied.

The Leiden community detection algorithm is highly similar to the more common Louvain detection algorithm but improves on it by guaranteeing well-connected communities[31]. These algorithms seek to maximize the modularity of the network, which is calculated as:

$$Q = \sum_{ij}\left(A_{ij} - \gamma k_i k_j/2m\right)\delta(\sigma_i, \sigma_j) \quad (6)$$

where $A$ is the adjacency matrix, $k_i$ is the weighted degree of node $i$, $m$ is the total sum of edge weights and $\delta\left(\sigma_i, \sigma_j\right) = 1$ if $i$ and $j$ are in the same community. $\gamma$ is the resolution parameter that defines the expected number of communities. A greater resolution coefficient results in more communities. In our dataset, the optimal resolution factor was derived empirically and set to 0.8. This value is motivated in Supplementary Notes 2 and Supplementary Fig. 2.

It was noted that if a protein contained regions of highly varying peptide densities, the algorithm tended to split clusters into high-density regions even though the peptides overlapped to a great extent. To correct this, a subsequent step was applied, where clusters were merged if the distance of each cluster termini was below the given threshold. Here, a threshold of 2 was used. Lastly, clusters with fewer than 3 peptides were removed from the dataset since these could not be quantified.

### Analyzing peptide clusters

The clusters can be quantified using different methods. We implemented two such methods, one in which it is quantified by the N most intense peptides, and one in which the N peptides exhibiting the largest difference metric, by DE score[40], fold-change or $p$ value, are used for quantification[41]. To investigate what clusters differed the most between the sample types, differential abundance analysis, and machine learning-based feature extraction were performed. For pairwise differential abundance analysis, $p$ values were calculated using

linear regression, and $Q$ values were computed by correcting for multiple hypothesis testing using the Benjamini-Hochberg procedure. When comparing more than two groups, $p$ values were calculated using one-way ANOVA.

To investigate if classification models could be used to distinguish between sample types, classification was performed with an XGBoost classifier. The classifier takes the z-scaled cluster intensities as input. The configuration for the XGBoost classifier was set to the default as per the implementation of the xgboost Python package. To investigate how many peptide clusters and which peptide clusters were important for classification, a modified recursive feature extraction scheme was implemented. Here, the feature matrix is iteratively reduced by removing the cluster with the lowest importance to the model. This iterative reduction of the feature matrix is conducted until a single feature remains. The feature importance is estimated using SHAP and is computed using the TreeExplainer as implemented in the SHAP Python package.

It was hypothesized that the high-dimensional cluster space could be used to estimate the level of bacterial colonization in the double-infected wounds. Weighted kNN ($k = 30$) regression was used to compute the cluster peptidomic likeness of the double-infected samples to the samples infected by $S. aureus$ and $P. aeruginosa$. The output was then correlated to the CFU fraction of $S. aureus$. Bootstrapping (bootstrap $N = 10$) was implemented by adding noise $\sim N(0,1)$ to the scaled features.

The most probable cut site giving rise to each cluster is the most common terminal, i.e., the mode of the terminal positions for each cluster and sample. This has been motivated in Supplementary Notes 3 and Supplementary Fig. 3. To identify cut site specificity, windows spanning 8 amino acids surrounding these sites were considered (p4-p4'). The amino acid distribution for each sample type and position was weighted with the mean peptide intensity in each cluster. The influence of the pathogens on the cut sites was compared against the control. To quantify the dissimilarity between these distributions, the KL divergence was computed as follows:

$$D_{KL} = \sum_{x \in X} p(x) \log\left(\frac{p(x)}{q(x)}\right) \tag{7}$$

Where $p(x)$ is the amino acid distribution for the sample type of interest and $q(x)$ for the sample type used as a comparison/background. For the single infections and double-infected samples, the control samples were used as the background distribution. The contaminated samples were compared against the $S. aureus$-infected samples, since this was the expected bacterial colonizer. The divergence was used to weigh the amino acid frequency at each position, before displaying them in a logoplot.

The cut site amino acid frequency distribution is a matrix of dimensions $23 \times 8$. Stacking this matrix into a vector of length $23 \cdot 8$ and concatenating all samples results in a matrix of size $N_{samples} \times 184$. To investigate the proximity of cut sites the dimensionality of the matrix was reduced to two dimensions using UMAP. The UMAP was configured as per the default configuration implemented in the umap-learn Python package.

### Comparison between peptide-centric and cluster-centric workflows on the urine peptidomes of type 1 diabetics

To investigate the generalizability of our method, we utilized a publicly available dataset from a study by Van et al.[37], which we downloaded from ProteomeXchange (PXD012210). The dataset consists of mass spectrometry (MS/MS) peptidomic data from 30 urine samples, including 15 samples from patients with type 1 diabetes and 15 control samples. Parts of the original analysis by Van et al. were re-created, specifically Fig. 2e, f in their publication. $P$ values were calculated using

linear regression and corrected for multiple hypothesis testing using the Benjamini–Hochberg method.

Clustering was performed with a resolution of 0.8 and a cutoff of 4, consistent with parameters used in other analyses within our study. Clusters were quantified using the top three peptides with the highest DE scores.

The feature quality improvement was examined through an iterative scheme. The original peptide and cluster feature matrices were subsetted to include the top N features sorted by $p$ value, starting with $N = 3$. The number of missing values for progressively larger feature matrices was identified. Additionally, we used the feature matrices to train 10 logistic regression classifiers on different stratified subsets of half the dataset and evaluated AUROC on the remaining halves. The logistic regression models were configured according to the defaults of sklearn 1.3.0. This iterative process continued until all features were included.

### Statistics and reproducibility

To ensure that the findings were reproducible, a random subset of samples was re-analyzed. These samples were chosen in a stratified but randomized manner from both singly infected samples and uninfected controls collected on day 1, with four samples selected from each group. The entire sample preparation pipeline and mass spectrometry analysis were repeated, but this time, the sample annotations were blinded. The annotations were revealed after creating clusters and projecting the cluster quantities to two dimensions using UMAP. Since the analysis of the original samples, the mass spectrometry park has been upgraded from timsTOF pro to timsTOF HT.

For zymogram analysis of porcine wound fluid samples from day 2; $n = 2$, for other wound fluid samples; $n = 1$. For zymogram analysis of human wound fluid samples, $n = 2$.

### Implementation and software

The complete bioinformatic analysis was conducted in Python 3.9. networkx and igraph were used to generate networks. The Leiden algorithm was used as implemented in the leidenalg-package. xgboost, scikit-learn, and shap were used for machine learning applications. The umap-learn package was used for dimensionality reduction. The processing package DPKS was used for quantification and differential abundance analysis[42]. A package for the creation of peptide clusters is available under an MIT license[24]. All illustrations were created in BioRender.com.

### Reporting summary

Further information on research design is available in the Nature Portfolio Reporting Summary linked to this article.

## Data availability

The raw mass spectrometry data for both the porcine and human samples has been deposited to proteomeXchange under the identifier PXD048892. The dataset from Van et al. was fetched from proteomeXchange with the dataset identifier PXD012210[37]. Source data are provided with this paper.

## Code availability

A package containing the code used for the creation and analysis of peptide clusters is available in the open GitHub repository https://github.com/ErikHartman/pepnets under an MIT license[24]. This repository also contains the notebooks which allow for the re-creation of all figures.

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

## Acknowledgements

We thank the Swedish National Infrastructure for Biological Mass Spectrometry (BioMS) for performing the LC-MS/MS analysis. Xinnate AB provided the funding and project management resources enabling the clinical safety study on non-healing wounds that generated the control samples used in this work. We are indebted to Drs. Sigrid Lundgren, Karl Wallblom, and colleagues at the Department of

Dermatology and the Clinical Trial Unit at Skane University Hospital Lund for patient-related work and for providing the patient wound fluids used in this study and to Dr. Ganna Petruk and Ann-Charlotte Strömdahl and José Cardoso for support with the wound fluid preparations. We thank Dr. Bo Nilson, Division of Medical Microbiology, Department of Laboratory Medicine, Lund University, for the identification of bacteria isolated from non-healing ulcers. We acknowledge support by grants from the Swedish Research Council (projects 2017-02341, 2018-05916 and 2020-02016 (A.S.), and 2023-02107 (J.M.)), Edvard Welanders Stiftelse and Finsenstiftelsen (Hudfonden) (A.S. and M.P.), the Royal Physiographic Society (A.S.), the Crafoord Foundation (M.P.), the Österlund Foundation (A.S.), and the Swedish Government Funds for Clinical Research (ALF) (A.S.).

## Author contributions

E.H. and A. Schmidtchen conceptualized the manuscript. E.H., J.M., J.P., and A. Schmidtchen designed the methodology. F.F., S.K., J.P., C.L. and M.P. performed the laboratory experiments. M.P. and A. Schmidtchen provided the porcine and patient samples. F.F. and S.K. performed sample preparation and mass spectrometry analysis. E.H. designed and performed the computational analysis and made the figures. E.H. wrote the initial manuscript with input from F.F., S.K., J.P., C.L., A. Schmidtchen, M.P., J.M. and A. Scott. All authors have read and agreed to the published version of the manuscript.

## Funding

## Competing interests

A Schmidtchen is a founder of in2cure AB, a parent company of Xinnate AB, which sponsored the clinical trial from which the biobank samples used in this study are derived. The authors declare no competing interests.
