## [Transparent Peer Review file · Nature Communications]

Peptide clustering enhances large-scale analyses and reveals proteolytic signatures in mass spectrometry data

Corresponding Author: Mr Erik Hartman

Figures originally included in the author's rebuttal have been redacted from this file.

Version 0:

Reviewer comments:

Reviewer #1

(Remarks to the Author)

This study presents an innovative computational workflow that leverages inherent peptide clustering to analyze complex peptidomic data. The application to wound fluid analysis and pathogen-specific colonization identification is particularly intriguing. My critique focuses solely on the computational strategies employed, as I do not have expertise in the experimental methods employed. While the work holds promise, I have several concerns regarding the computational methodology:

1. The choice of the optimal community detection value (4) needs clearer justification within the manuscript. It would be helpful to describe the process or criteria used to determine this value.
2. The manuscript would benefit from a more detailed explanation of the SVM and XGBoost model training procedures. This includes Feature optimization methodology, Parameter search ranges used during model development, and Training dataset size and composition.
3. A discussion on why SVM and XGBoost were selected over deep learning-based classifiers would provide valuable insight into the methodological decision-making.
4. Clarification is needed on whether any parameter optimization was performed to achieve the clear clustering observed in the UMAP visualization (Figure 5). A brief explanation of this process would be beneficial.
5. While the authors address the importance of peptide length consistency, a quantitative comparison of performance improvement (handling peptides of similar length vs. varying lengths) would strengthen the findings.

Reviewer #2

(Remarks to the Author)

Currently, the analysis of peptidomic data resulting from biological or clinical samples suffers from large amounts of missing values in quantitative information. This problem causes that the number of quantifiable peptides across large sample cohorts is limited after filtering out the unreliable peptides that contain a certain proportion of missing values. To solve this problem, the authors in this manuscript developed a community-based peptidomic clustering strategy, where the peptides were clustered according to the similarity and proximity in the overlapping sequences to reduce the dimensionality of peptidomic data. This strategy was validated using a set of peptidomic data resulting from wound infection animal models as well as the clinical samples collected from patients with non-healing wounds. Overall, this manuscript is well-written and present certain novelty, and most of the conclusion can be supported by the results. However, this reviewer still has some concerns as shown below.

Major issues:

1. Comparison with the results obtained by a traditional peptidomic approach is needed.

In the first four sub-sections of Results (pages 2-10), the authors introduced the rationale of this new clustering strategy and also demonstrated the utility using a set of peptidomic data collected from wound fluid samples of infected porcine. The new clustering strategy enabled analysis of the peptidomic data and identified the bacterial protease activity. Three proteins with different degradation patterns dependent on infection type was also illustrated. This reviewer appreciates the efficiency of

this new strategy developed by authors. However, without comparison with a negative control using traditional data processing approach, it is less confident to draw these conclusions. Thus, this reviewer suggests that a previous reported peptidomic approach needed to be used for a comparison, where the unreliable peptides with certain missing values are filtered out and after imputation, the peptidomic data is quantified (as reported in PMID: 34227939, 36266275 or other previous publications). The authors should demonstrate that the data shown Figures 2-4 cannot be obtained by using the traditional peptidomic approach, or the proposed new clustering strategy can generate more accurate classifiers than the traditional one.

In addition, analysis of a publicly available peptidomic dataset published by other groups may be helpful to demonstrate the utility and efficacy of this new strategy and exclude the possible analytical biases.

2. The top-N peptides in each cluster may be phenotype-specific, as well.

In page 5, the authors quantified the cluster by taking of the sum of the three most abundant peptides (top-N method). This reviewer speculates that the top-N peptides in each cluster could pass the threshold for filtering out unquantifiable peptides in traditional peptidomic approach, since high abundant peptides usually contain fewer missing values across samples. Perhaps, without clustering, just use of the top-N peptides could give rise to the similar results in Figure 2b and 2d. Please prove this speculation is wrong. Otherwise, this clustering strategy is less significant.

3. The clinical application needs to be refined.

One of the applications for clinical proteomics or peptidomics is to identify protein or peptide makers for early diagnosis, therapy or prognosis of diseases. The authors claim that the new strategy described in this manuscript can identify the cluster-based markers in diseases (in the last sentence of the abstract). This reviewer agrees that it represents an innovation in this field. However, the current results can only prove the feasibility of this new strategy, instead of identification of any disease markers. Perhaps, analysis of wound fluid peptidomics induced by bacterial infection is an improper application scenario for this strategy. As the authors stated, the qualitative bacterial composition of the wounds can be determined by MALDI, and the protease activity can be analyzed by zymograms. What is the unique information can be obtained from the cluster-based peptidomics strategy to facilitate the diagnosis or therapy? Even though the authors claimed to obtain pathogen-specific proteolytic activity, the current results and related statement still lack a close connection between the peptidomic data and clinical applications.

4. The description for predicting proteolytic cut-sites is unclear.

Although the cluster-based strategy is efficient for processing the peptidomic data, prediction of the proteolytic cut-sites is RISKY. One peptide cluster contains a large number of peptides with various length resulting from different proteolytic position in the precursor proteins. Which peptide's intensity can be used in the topN method? The current description in the last paragraph of page 7 is unclear and cannot convince this reviewer.

Minor issues:

1. In page 2, the sample numbers have been shown with *S. aureus* (N=21) and *P. aeruginosa* (N=17). Were the samples collected from the same animal, or one sample from each animal? How did the authors avoid possible batch effect as well as individual difference?
2. In page 3, except for peptide identifications, the protein identifications need to be shown, since it represents the precursors of these peptides.
3. In page 8, why only show the three proteins, HMGB1, HPT and PR-39? How many proteins in the results show pathogen-specific peptidomic clusters?
4. In page 9, there is no Figure 4d in this manuscript, even though mentioned in the text.
5. In page 11, Figure 5e is not mentioned in the text.
6. In page 12, the color key in Figure 5d is missing.
7. In page 13, the data of MALDI to determine the bacterial composition need to be shown in the supporting information.
8. In page 15, the authors discussed about the limitation of the dataset. To demonstrate this cluster-based strategy with large sample cohort, this reviewer suggests to take advantage of publicly available datasets from other research groups.
9. In the section of Methods, the "wet" experiments should be described prior to the "dry" one, since this research was conducted according to this sequential order.
10. In page 20, quantification settings are missing in the database search using PEAKS X engine. The date for downloading the protein database need to be clarified.
11. In this manuscript, the "wet" experiment contributes a lot, including the animal models, sample preparation, LC-MS analysis, etc. In the section of Authors' contributions, the person who performed "wet" experiment need to be mentioned, instead of just provision of samples.

Reviewer #3

(Remarks to the Author)

Hartman and colleagues submitted a manuscript entitled “Community-based peptidomic clustering reveals pathogen-specific proteolytic signatures in wounds” for publication. The authors report on the peptidome analysis of fluid collected from porcine wounds infected with *S. aureus* and/or *P. aeruginosa* and subsequent investigation of human wounds. In addition to the issues detailed below, this reviewer was struggling to understand the main aim of the manuscript, the purpose of the investigation. The authors indicate that it is possible to differentiate between *S. aureus* and *P. aeruginosa* infected wounds, but this obviously can also be achieved by other, much easier approaches. The (added) benefit if this quite extensive investigation is not evident from the manuscript.

Further specific issues:

Some statements need revisiting. For example the sentence “Current peptidomic data analysis strategies largely rely on filtering-methods to sift through the large datasets and remove unwanted degradation products to eventually identify some of the relevant bioactive peptides of importance” does not seem to be correct, based on the current literature.

The study appears substantially underpowered, especially in the light of the enormous variability observed. Apparently most peptides identified were observed in only one or very few samples (“On average, each sample contained $89.5 \pm 7\%$ missing values”), raising concerns about the validity of the peptide identification especially in cases where the peptide was detected only once. Reproducibility has apparently not been addressed at all.

The fact that most peptides are observed in one or very few cases may substantially contribute to the apparent efficiency of the dimensionality reduction, which may be simply due to focusing on peptides that are more frequently observed.

The low power of the study, the fact that it includes a very large number of peptides that were observed only once, also results in concerns about the subsequent data presented on the protease prediction and on the pathogen specific clusters.

Overall, the current manuscript reports on a substantially underpowered study, especially when considering the high variability. Further, the main aim of the study is unclear, it may be 1) presenting an approach for dimensionality reduction of peptidome data, or 2) presenting a value of peptidome analysis for wound management. In case of 1), the use of a much larger dataset (e.g. from the many available peptidome studies with a clinical focus) seems advisable to demonstrate a benefit of the approach, in case of 2) a much larger study that allows statistical analysis, with a clearly stated aim how to improve the current state-of-art and/or knowledge would be needed.

Reviewer #4

(Remarks to the Author)

Reviewer #5

(Remarks to the Author)

Author Rebuttal letter:

General comments:

We thank the reviewers and the editors for their insightful and constructive comments. After careful consideration of the feedback and substantial changes to the manuscript, we believe that our work has undergone significant enhancement, becoming more clear in its presentation and more compelling in its potential utility.

Our responses and amendments have been centered around three main aspects: clarity, utility, and reproducibility.

Clarity: Recognizing concerns raised by reviewers 1 & 2 about certain technical aspects of the clustering algorithm and downstream analysis methodologies, we have dedicated substantial effort to clarify these points by including illustrative examples and clearer explanations.

Utility: Reviewers 2 & 3 raised questions regarding the utility of our algorithm compared to traditional approaches of peptidomics analysis and other experimental techniques such as MALDI-TOF and zymograms. We have therefore made comparisons with traditional peptide-centric peptidomics approaches and these approaches to our own data and to a new publicly available dataset. These comparisons highlight the advantages and potential applications of our approach compared to the traditional peptide-centric peptidomics data analysis strategies. We have also clarified the advantages of using an MS-based peptidomics approach in the main text and figures.

Reproducibility: Questions regarding reproducibility were raised by reviewer 3, especially in light of the large amount of missing values. To address these, a stratified random subset of samples was re-analyzed blindly. The results demonstrate that the analysis was reproducible despite the challenges of missing values. We have also provided notebooks that can be used to re-create the entire analysis to ensure reproducibility.

With these amendments, all figures have been remade. In the process of doing so, we realized minor improvements to various elements of our analysis, such as proper handling and labeling of non-proteotypic peptides and clusters. These changes have altered the appearance of some plots, but have not made any difference to the overall conclusions of the paper.

A point-by-point response to the reviewers' concerns can be found below.

1 REVIEWER COMMENTS

Reviewer #1 (Remarks to the Author):

This study presents an innovative computational workflow that leverages inherent peptide clustering to analyze complex peptidomic data. The application to wound fluid analysis and pathogen-specific colonization identification is particularly intriguing. My critique focuses solely on the computational strategies employed, as I do not have expertise in the experimental methods employed. While the work holds promise, I have several concerns regarding the computational methodology:

1. The choice of the optimal community detection value (4) needs clearer justification within the manuscript. It would be helpful to describe the process or criteria used to determine this value.

We acknowledge the reviewers' valid critique regarding the lack of a comprehensive description of the parameter selection process crucial for peptide cluster creation. Furthermore, we recognize the necessity for elucidating the underlying algorithm, particularly in terms of parameter implications. In response, we have generated an additional figure, now included in the Supplement, alongside descriptions and clarifications in the Methods and Supplement, to address these concerns.

As discussed in the paper, devising an objective function proved challenging for objectively assessing optimal parameter values. Instead, we ramped the parameters (resolution and cutoff) and visually assessed the impact on the clustering. We evaluated the clustering based on the extent of split clusters and clusters that contained non-overlapping incorrect peptides as seen in the Figure below. In this analysis, a threshold cutoff of 4 and a resolution of 0.8 were the parameters that generated the most accurate results (center panel in d below). We also discovered that generating larger clusters reduced the dimensionality of the dataset but increased the number of false-negative cut-sites. We realize that tuning parameters without an objective function is suboptimal, however, the flexibility to change parameters permits on the other hand the users to tailor the clustering for their dataset and their scientific question.

[Redacted]

2
Supplementary Fig. 2 The impact of parameters on the clustering of peptidomes. a An illustration showcasing the impact of the edge distance cutoff used in peptide network creation. b An illustration showcasing the impact of the resolution parameter in peptide cluster partitioning with the Leiden algorithm. c The networks when varying the cutoff (1,5,10) were projected to 2 dimensions using uniform manifold approximation and projection (UMAP). Each node is a peptide in hemoglobin subunit β (HBA) and the node colors reflect the peptide midpoints. d Peptide networks for HBA were created with a cutoff of 4 and then partitioned with a varying resolution parameter (0.1, 0.8, 1.6). The nodes are colored based on their cluster designation. e Peptide projections when varying the resolution (y-direction) and cutoff (x-direction). The peptides are colored based on their cluster designation. The middle panel shows a resolution of 0.8 and a cutoff of 4, which were the parameter values used in the study.

3
2. The manuscript would benefit from a more detailed explanation of the SVM and XGBoost model training procedures. This includes Feature optimization methodology, Parameter search ranges used during model

development, and Training dataset size and composition.

3. A discussion on why SVM and XGBoost were selected over deep learning-based classifiers would provide valuable insight into methodological decision-making.

Combined response for points 2 and 3: We appreciate these comments, as they made us re-evaluate the goals of the machine learning (ML) methods used in the manuscript, and how they were communicated to the reader. As a result, we have amended the methods and results sections, and Figure 2.

In our manuscript, we addressed three aims with the ML-based classification: 1) How many clusters have unique and useful information for classification, and to what extent can we reduce the dataset without losing classification accuracy? 2) How accurately can the type of wound infection be classified? 3) Can ML be used to determine the relative number of bacteria in super-infected wounds? These aims are addressed in Fig. 2e, f, and g respectively.

The reasoning behind employing XGB when evaluating 1) was that the TreeExplainer in SHAP is efficient, and XGB is a good boosted-tree implementation. We wanted to use the same model in points 1) and 2), hence the model choice for 2). A vanilla XGBClassifier, as implemented in the Python package xgboost (https://xgboost.readthedocs.io/en/stable/python/python_api.html#xgboost.XGBClassifier), achieved high cross-validation accuracy, and no changes were made to the model. The recursive feature elimination process using SHAP and the model evaluation has been clarified in the method section.

We do agree that DL has great potential in discovery-peptidomics applications like this, however, the simpler models used here were sufficient for our applications and data. Given that we reached nearly perfect accuracy with an XGBoost model, we see no reason to make a DL model.

For aim 3 above, we originally employed an SVC with a linear kernel, as it has a linear decision function that underfitted the data. We speculated that this compromise made by the model would reflect the relative bacterial count (CFU) in the super-infected wounds. When addressing the reviewer's comments, we realized that a more elegant way to address this aim would be to use kNN regression. Here, the regressor was made to predict the type of bacteria in the wounds (0 = *P. aeruginosa*, 1 = *S. aureus*). We have also added bootstrapping (bootstrap $N=10$), where noise $\sim N(0,1)$ was added to the scaled feature values before running kNN. We used $k=30$ which corresponds to about $\sim 50\%$ of the training samples, but the results are consistent for other values of k . This results in a Pearson r coefficient of 0.93 ± 0.03 and is accomplished with a more intuitive method. We have replaced the old panel with these results. We have also added a schematic illustration of the approach in Fig. 2 for clarity.

[Redacted]

4. Clarification is needed on whether any parameter optimization was performed to achieve the clear clustering observed in the UMAP visualization (Figure 5). A brief explanation of this process would be beneficial.

4

For all UMAPs, the default values in umap-learn (<https://umap-learn.readthedocs.io/en/latest/>) were used. The only modification to the data was scaling by z-score. This has been clarified in the manuscript.

5. While the authors address the importance of peptide length consistency, a quantitative comparison of performance improvement (handling peptides of similar lengths vs. varying lengths) would strengthen the findings.

The distance function was created in an iterative process during which unsatisfactory clusters were identified and solutions were developed to deal with these issues. For the length consistency term, this fixed an issue where subpeptides were mapped onto the cluster of the parent peptide, even though they largely differed in length. Because this difference in length was unlikely to be the result of exoprotease activity but instead a new endoprotease cut, our goal was to separate these peptides into different clusters. An example of such a case in HPT, with and without the length consistency term, is shown below. We were not able to define a quantifiable objective to measure of clustering performance. However, we have added text to the method section to clarify individual terms in the distance function used when creating networks prior to community detection. In addition, we realized that it could be beneficial for other users to add coefficients for each term so that users can adjust these individually. So, we have modified the function in the companion Python-package and updated the methods section accordingly.

[Redacted]

Currently, the analysis of peptidomic data resulting from biological or clinical samples suffers from large amounts of missing values in quantitative information. This problem causes that the number of quantifiable peptides across large sample cohorts is limited after filtering out the unreliable peptides that contain a certain proportion of missing values. To solve this problem, the authors in this manuscript developed a community-based peptidomic clustering strategy, where the peptides were clustered according to the similarity and proximity in the overlapping sequences to reduce the dimensionality of peptidomic data. This strategy was validated using a set of peptidomic data resulting from wound infection animal models as well as the clinical samples collected from patients with non-healing wounds. Overall, this manuscript is well-written and present certain novelty, and most of the conclusion can be supported by the results. However, this reviewer still has some concerns as shown below.

Major issues:

1. Comparison with the results obtained by a traditional peptidomic approach is needed.

In the first four sub-sections of Results (pages 2-10), the authors introduced the rationale of this new clustering strategy and also demonstrated the utility using a set of peptidomic data collected from wound fluid samples of infected porcine. The new clustering strategy enabled analysis of the peptidomic data and identified the bacterial protease activity. Three proteins with different degradation patterns dependent on infection type was also illustrated. This reviewer appreciates the efficiency of this new strategy developed by authors. However, without comparison with a negative control using traditional data processing approach, it is less confident to draw these conclusions. Thus, this reviewer suggests that a previous reported peptidomic approach needed to be used for a comparison, where the unreliable peptides with certain missing values are filtered out and after imputation, the peptidomic data is quantified (as reported in PMID: 34227939, 36266275 or other previous publications). The authors should demonstrate that the data shown Figures 2-4 cannot be obtained by using the traditional peptidomic approach, or the proposed new clustering strategy can generate more accurate classifiers than the traditional one.

In addition, analysis of a publicly available peptidomic dataset published by other groups may be helpful to demonstrate the utility and efficacy of this new strategy and exclude the possible analytical biases.

We have addressed these concerns with two main actions:

1. We have analyzed a previously published dataset from Van et al. (Peptidomic Analysis of Urine from Youths with Early Type 1 Diabetes Reveals Novel Bioactivity of Uromodulin Peptides In Vitro, MCP, 2020, PMID: 31879271, ProteomeXchange ID: PXD012210).
2. We have analyzed our own data with a traditional peptidomics approach and highlighted some key improvements made possible with our algorithm.

1: The search results from Van et al. were downloaded from ProteomeXchange. We recreated their analysis using the workflow from their manuscript, as schematically shown in a below. The key finding in their study was the identification of 7 peptides derived from Uromodulin (UMOD). We were able to reproduce their results by differential abundance analysis after filtering out peptides with missing values (b,c below). We then made a similar analysis but with the key difference of creating clusters. We implemented a new quantification method that quantifies the clusters based on the top N peptides defined by their differential expression score (DE-score). The DE-score is a quantitative measure of differential abundance, taking both fold change and Q-value into consideration. The differential Uromodulin peptides are all found within one cluster and were easily identified by our method. Importantly, we also identify 2 additional clusters that were overlooked in the original paper, that passed the Q-value threshold (<0.05). These peptide clusters are from lectin mannose binding protein 2 (LMAN2) and secreted and transmembrane protein 1 (SECTM1). The figure below illustrates the comparison and has been added to the Supplement Material. Interestingly, we also discovered that the peptide-cluster clustermap showed a clear separation into two possible subtypes of type 1 diabetes, which was not as apparent when clustering on the peptides alone. These results could not be obtained by using the traditional peptidomic approach and shows the utility of the proposed new clustering strategy.

[Redacted]

Supplementary Fig. 3 Comparison with a traditional peptidomic approach on a publicly available dataset. a The results of our re-creation of the workflow presented in Van et al. b Volcano plot corresponding to Fig. 2e in Van et al., except that we show the Q-value on the y-axis instead of the p-value. c Clustermap recreating Fig. 2f in Van et al. d The results of performing a similar workflow but with the creation of peptide clusters. Clusters were quantified with the top 3 most differentially abundant peptides. e Volcano plot similar to in b. f Clustermap of the top 20 clusters by p-value. All significant clusters were not included for ease of visualization.

2: To compare a traditional workflow with the new clustering-based approach using our data, we analyzed our

data using a workflow similar to the one presented in Van et al. (2020). For simplification, comparisons were made on the differentially abundant peptides when comparing the wounds infected by *P. aeruginosa* and *S. aureus*. Similarly to Van et al., we apply a filter and only keep peptides with evidence in more than 14 samples per group. In the traditional peptidomics approach, this includes 652 peptides thereby discarding 14247 peptides. This is in contrast with the clustering shown here which includes 5237 peptides. The figure below showcases some key differences in the results and has been added to the Supplement section.

First, the traditional peptidomic approach generated considerable redundancy regarding biomarkers/differentially abundant peptides. Many of the differentially abundant peptides were derived from the same protein region with overlapping peptide sequences and highly correlated abundance profiles. When clustering all peptides prior to statistical testing, a more concise and comprehensible picture emerges that enhances the interpretation of the results as shown in a, b below. Second, we demonstrate that the clustering approach improves the cut site analysis. The primary challenge posed by conventional methods predominantly pertains to noise. We speculate that many of the identified peptides in peptidomics data are a result of unspecific exoprotease activity, and these false positives mask the specific true positive endoprotease activity. In our approach, the clustering would remove some of the false positives in cut-site detection, thereby removing some of the noise from the signal, as illustrated in c below. Here, the Kullback-Leibler divergence was greater when computing the specificity using peptide clusters indicating that this is the case (d below). This information was added to the Supplement Material and has also been highlighted in Fig. 3a.

[Redacted]

7

Supplementary Fig. 4 Comparison between clustering and traditional analyses. a The clustering approach entails creating clusters and quantifying them. Here we quantified the clusters with the top 3 most differentially abundant peptides. The top panel shows a volcano plot. The Q-value is calculated by computing a p-value using linear regression followed by Benjamini-Hochberg correction for multiple hypothesis testing. The bottom panel shows a clustermap of the top 20 clusters by Q-value. b Similar to a but with a traditional peptidomics approach. In the clustermap, it is apparent that many peptides are highly similar and highly correlated. m denotes modification. c An illustration of the benefits of using clusters for cut site identification. The sharper distribution is a result of removing the false positive cut sites introduced by exoproteases and leads to a larger KL divergence. d Logoplots showcasing the p4-p4' cut sites when using a clustering approach (left) vs. a traditional approach (right).

2. The top-N peptides in each cluster may be phenotype-specific, as well.

In page 5, the authors quantified the cluster by taking of the sum of the three most abundant peptides (top-N method). This reviewer speculates that the top-N peptides in each cluster could pass the threshold for filtering out unquantifiable peptides in traditional peptidomic approach, since high abundant peptides usually contain fewer missing values across samples. Perhaps, without clustering, just use of the top-N peptides could give rise to the similar results in Figure 2b and 2d. Please prove this speculation is wrong. Otherwise, this clustering strategy is less significant.

To address this comment, we filtered the peptide list to contain the top 5% most intense peptides and can confirm that these peptides are not the same peptide list as the top-N peptides used to quantify the clusters. Interestingly, the most intense peptides are only derived from 48 proteins, whereas the clusters are derived from 140 proteins. Producing a UMAP as in 2b with these peptides results in worse separation, as shown below.

[Redacted]

8

Further, when extracting the most differentially abundant peptides, as in 2d, we identify that several of these are derived from the same protein regions and therefore provide less information, as shown below.

[Redacted]

Interestingly, the concern raised by the reviewer prompted us to explore alternative strategies the top-N method to quantify the peptide clusters, as elaborated in our response to point 1. In this alternative strategy, the clusters were quantified based on peptides exhibiting the highest DE-score rather than the most intense ones. Consequently, we have also implemented additional methods, such as quantifying based on fold change and Q-value in the Python package to diversify the range of quantification methods. This underscores that while clustering serves to unveil relationships between peptides, the subsequent quantification method remains adaptable and can be tailored to meet the specific requirements of a given dataset or study.

3. The clinical application needs to be refined.

One of the applications for clinical proteomics or peptidomics is to identify protein or peptide makers for early

diagnosis, therapy or prognosis of diseases. The authors claim that the new strategy described in this manuscript can identify the cluster-based markers in diseases (in the last sentence of the abstract). This reviewer agrees that it represents an innovation in this field. However, the current results can only prove the feasibility of this new strategy, instead of identification of any disease markers. Perhaps, analysis of wound fluid peptidomics induced by bacterial infection is an improper application scenario for this strategy. As the authors stated, the qualitative

9

bacterial composition of the wounds can be determined by MALDI, and the protease activity can be analyzed by zymograms. What is the unique information can be obtained from the cluster-based peptidomics strategy to facilitate the diagnosis or therapy? Even though the authors claimed to obtain pathogen-specific proteolytic activity, the current results and related statement still lack a close connection between the peptidomic data and clinical applications.

We thank the reviewer for their insightful evaluation and for recognizing the innovative nature of our work. We also recognize that the previous version of our manuscript, while providing an overview of the wound fluid peptidome from a peptide cluster perspective, failed to provide concrete potential biomarkers or likewise elucidate the process one would go about in doing so. Prompted by this and other points raised by the reviewer, we have therefore run new analyses and altered several figures.

Importantly, in the context of wound fluid peptidomics and bacterial infection, our strategy identifies clusters of peptides that are specifically altered in response to different bacterial pathogens, as exemplified in the new Figures 4 & 5, in which we highlight HBA 0-16 as a potent biomarker for early detection of *P. aeruginosa* which becomes differentially abundant as soon as *P. aeruginosa* is the primary colonizer. We also demonstrate that the peptide clusters can be used to assess the relative bacterial CFU in superinfected wounds, as highlighted in Fig. 2g and the response to reviewer 1. Hence, these data provide solid proof of principle demonstrating the application of the methodology in infected wounds.

We still concur with the assessment that our study primarily demonstrates the feasibility of our new strategy, rather than the identification of specific clinically validated disease markers. The unique aspect of our cluster-based peptidomics strategy is its potential to provide a comprehensive view of the peptidome through dimensionality reduction of mass spectrometry data. Accordingly, to emphasize the conceptual nature of our work, we have changed the title to "Peptide clusters reveal proteolytic signatures and biomarkers in mass spectrometry data". While we have not yet clinically validated specific disease markers, we believe that our approach represents a significant step forward in the field of peptidomics, providing a new analytical tool of value for future large-scale clinical studies.

Although these results are obtainable by MALDI-TOF analysis this process is slower and selective. MALDI-TOF analyses are based on bacterial colonies that are selected and picked from plates after overnight culturing, which is time-consuming and selective in that not all bacteria can be cultivated easily. Importantly, MALDI-TOF analysis does not provide functional data on the bacteria's ability to induce endogenous inflammatory pathways or release proteases. This information is captured in our approach in addition to the assessment of relative bacterial CFU in vivo, and the cut site specificity. Such information is valuable when trying to understand the underlying biology of infection, and as a starting point to identify disease-relevant biomarkers. Likewise, zymography is also selective and can only detect gelatinase activity. This is a considerable limitation and cannot detect the activity of e.g. human neutrophil elastase and multiple other enzymes. Although the human material in this study is limited, we have included clusters that represent a phenotype not identifiable by MALDI-TOF or zymograms. This information was included in the revised Fig. 6 and highlights the added information provided by an MS-based approach.

Lastly, we would like to point out that aim of this work was to introduce peptide clustering into peptidomics. The approach is generic and the algorithm we developed is freely available and can be applied to larger clinical studies, where it could facilitate novel insights into disease mechanisms and the discovery of new biomarkers for other diseases.

4. The description for predicting proteolytic cut-sites is unclear.

Although the cluster-based strategy is efficient for processing the peptidomic data, prediction of the proteolytic cut-sites is RISKY. One peptide cluster contains a large number of peptides with various length resulting from different proteolytic position in the precursor proteins. Which peptide's intensity can be used in the topN method? The current description in the last paragraph of page 7 is unclear and cannot convince this reviewer.

We have partly addressed this in the comment under point 1 above. As seen in the response under point 1, the primary advantage of utilizing clusters for proteolytic cut site prediction lies in enhancing the signal-to-noise ratio. We believe that the reason for this is the combined activity of both endoproteases and exoproteases, where endoprotease cuts are considered a signal due to their specificity, and exoprotease cuts introduce noise. In this case, the clustering enhances the proteolytic position in the precursor protein. We would like to point out that

10

there is a discernable signal even without clustering, which, although dampened, shows a similar profile as when considering cut sites. We have added an illustration to Fig. 3 in the manuscript which showcases our reasoning for the reader. Furthermore, we have provided a better explanation in the method section and the Supplement which describes the algorithm and advantages.

Minor issues:

1. In page 2, the sample numbers have been shown with *S. aureus* (N=21) and *P. aeruginosa* (N=17). Were the samples collected from the same animal, or one sample from each animal? How did the authors avoid possible batch effect as well as individual difference?

The samples were collected from 12 different pigs as indicated in Fig. 1a. We have now added a Supplement table to the Supplement Material containing information about the origin of all samples including pig and wound IDs.

To mitigate possible batch effects and technical variations and to prevent carryover during data acquisition, all samples were analyzed in one MS batch with blanks between every sample. In the post-processing, the intensities were mean-normalized per sample. As shown in the UMAPs below, the samples do not cluster on pig ID but rather cluster on time point and infection type indicating that the steps taken to minimize batch effect were sufficient.

[Redacted]

2. In page 3, except for peptide identifications, the protein identifications need to be shown, since it represents the precursors of these peptides.

This has been corrected.

3. In page 8, why only show the three proteins, HMGB1, HPT and PR-39? How many proteins in the results show pathogen-specific peptidomic clusters?

This point made us remake Fig. 4 to show a clearer description of how differentially abundant clusters can be identified. We still show three examples, which were chosen to highlight biologically interesting clusters. During the re-analysis of the data, we realized that HBA (0-16) is a highly interesting cluster and exchanged HPT with HBA. This cluster is located in the N-terminal site of the protein and a cut-site with lysine at p1 on the C-terminal, and is consistently highly abundant in wounds infected by *P. aeruginosa*.

The 68 clusters that exhibit a |fold change| > 2 and a Q-value < 0.05 are from 33 unique proteins.

4. In page 9, there is no Figure 4d in this manuscript, even though mentioned in the text.

This has been corrected.

5. In page 11, Figure 5e is not mentioned in the text.

11

All panels are now mentioned in the text.

6. In page 12, the color key in Figure 5d is missing.

This has been added.

7. In page 13, the data of MALDI to determine the bacterial composition need to be shown in the supporting information.

We have added this information.

8. In page 15, the authors discussed about the limitation of the dataset. To demonstrate this cluster-based strategy with large sample cohort, this reviewer suggests to take advantage of publicly available datasets from other research groups.

This is a good idea. We have analyzed previously published dataset from Van et al. (Peptidomic Analysis of Urine from Youths with Early Type 1 Diabetes Reveals Novel Bioactivity of Uromodulin Peptides In Vitro, MCP, 2020,

PMID: 31879271, ProteomeXchange ID: PXD012210). The results obtained from this analysis are shown under point 1 above and the information has been added to the manuscript.

9. In the section of Methods, the wet experiments should be described prior to the dry one, since this research was conducted according to this sequential order.

The order has been changed.

10. In page 20, quantification settings are missing in the database search using PEAKS X engine. The date for downloading the protein database need to be clarified.

These have been added.

11. In this manuscript, the wet experiment contributes a lot, including the animal models, sample preparation, LC-MS analysis, etc. In the section of Authors' contributions, the person who performed wet experiment need to be mentioned, instead of just provision of samples.

We have clarified this in the Authors' contribution section.

12

Reviewer #3 (Remarks to the Author):

Hartman and colleagues submitted a manuscript entitled "Community-based peptidomic clustering reveals pathogen-specific proteolytic signatures in wounds" for publication. The authors report on the peptidome analysis of fluid collected from porcine wounds infected with *S. aureus* and/or *P. aeruginosa* and subsequent investigation of human wounds.

In addition to the issues detailed below, this reviewer was struggling to understand the main aim of the manuscript, the purpose of the investigation. The authors indicate that it is possible to differentiate between *S. aureus* and *P. aeruginosa* infected wounds, but this obviously can also be achieved by other, much easier approaches. The (added) benefit if this quite extensive investigation is not evident from the manuscript.

We apologize for the lack of clarity. Our study has two main aims. The first aim is to introduce the concept of peptide clusters and to provide an open-source algorithm to enable others to use the approach as an alternative to traditional peptidomic data analysis strategies. The second aim is to showcase the utility of our presented concept by analyzing infected wounds to gain new insight into the biology underlying the influence of *P. aeruginosa* and *S. aureus* on the proteolytic environments. We have made substantial changes to the manuscript to make these aims are more clear. Firstly, we have elucidated the algorithm and its advantages to address aim 1. Secondly, the analysis of the wound fluids has been enhanced to address aim 2.

Further specific issues:

Some statements need revisiting. For example the sentence "Current peptidomic data analysis strategies largely rely on filtering-methods to sift through the large datasets and remove unwanted degradation products to eventually identify some of the relevant bioactive peptides of importance" does not seem to be correct, based on the current literature.

The sentence above has been reworded to: "Current peptidomic analysis strategies largely rely on computational methods to sift through the large datasets to identify relevant bioactive peptides."

The study appears substantially underpowered, especially in the light of the enormous variability observed. Apparently most peptides identified were observed in only one or very few samples ("On average, each sample contained $89.5 \pm 7\%$ missing values"), raising concerns about the validity of the peptide identification especially in cases where the peptide was detected only once. Reproducibility has apparently not been addressed at all.

We realize that the previous version of the text was not formulated clearly. One main point of introducing peptide clusters shown in our manuscript was to decrease the number of missing values. Our approach did in fact decrease the number of missing values by 300%. Furthermore, the clustering drastically reduces the number of peptides found in only one sample. In the porcine data, >3000 peptides were identified in a single sample. The corresponding number of clusters identified in a single sample was 8, a more than 35-fold decrease in single sample identifications. The reduction was even higher in the human dataset (see table below).

[Redacted]

The reviewer is correct in pointing out that we had not addressed reproducibility. To address this, we performed a

blinded re-analysis of 12 samples. These samples were picked stratified but randomly from the singly infected samples and the uninfected controls from day 1. The entire sample preparation pipeline and mass-spectrometry analysis were repeated but this time the sample annotations were blinded. We then analyzed the samples using the computational workflow and visualized the results using UMAPs to determine reproducibility (see the figures below). Encouragingly, the sample types are located in the correct infection and temporal clusters. Since the analysis of our original samples, we have upgraded our mass spectrometry park from timsTOF pro to timsTOF HT. The timsTOF HT is slightly different and has faster electronics, which certainly introduces batch effects. Still, the replicates cluster with the infection type and time point.

[Redacted]

13

Additionally, we show that the specific signal is reproduced as exemplified by the cluster intensity for HBA (0-16) below. The results demonstrate that the entire workflow from sample preparation, MS analysis, and computational workflow generates reproducible results.

Lastly, we would like to point out that our study has a larger number of samples used in most peptidomics studies (Fig. 1c in Madsen et al. 2022, Nat. Comms. Combining mass spectrometry and machine learning to discover bioactive peptides shows the size of representative peptidomics datasets). Furthermore, we have taken several measures to mitigate for the size of the sample cohort such as performing k-fold cross-validation, bootstrapping, and performing statistical corrections according to the standards of the field.

The fact that most peptides are observed in one or very few cases may substantially contribute to the apparent efficiency of the dimensionality reduction, which may be simply due to focusing on peptides that are more frequently observed. The low power of the study, the fact that it includes a very large number of peptides that were observed only once, also results in concerns about the subsequent data presented on the protease prediction and on the pathogen specific clusters.

We apologize for the lack of clarity in the original version of our manuscript. The point of introducing peptide clusters was to mitigate the issues highlighted above. Consequently, we did not use the peptides for dimensionality reduction and the clustering reduced the number of single sample identifications from >3000 peptides to 8 clusters. As shown in the response to reviewer 2 under point 1, the clustering also improves the definition of cut-sites. We have made substantial changes to the manuscript to clarify these concerns. Foremost, we have included clearer explanations of the methods and results, alongside illustrative schematics in the figures, e.g. Fig. 2, 3, and 4.

Overall, the current manuscript reports on a substantially underpowered study, especially when considering the high variability. Further, the main aim of the study is unclear, it may be 1) presenting an approach for dimensionality reduction of peptidome data, or 2) presenting a value of peptidome analysis for wound management. In case of 1), the use of a much larger dataset (e.g. from the many available peptidome studies with a clinical focus) seems advisable to demonstrate a benefit of the approach, in case of 2) a much larger study that allows statistical analysis, with a clearly stated aim how to improve the current state-of-art and/or knowledge would be needed.

14

As mentioned above, we have changed the text to clarify the main messages of the manuscript and to make it clearer that one of the main goals was to reduce the problem of missing values by peptide clustering. In addition, we have reanalyzed our manuscript using the traditional peptidomics workflow to show the added benefit of our approach (see points 1 and 2 in response to reviewer 2). We have also reanalyzed another previously published data set using the traditional peptidomics and our approach to demonstrate that our method can uncover findings that were beyond the reach of the traditional peptidomics approach. All of these analyses highlight the utility of our method by increasing the interpretability of large-scale peptidomics data and have been added to the Supplementary Material. Lastly, we have reanalyzed the data in several ways to concretize the concept of peptide clusters as biomarkers. In this process, we discovered HBA 0-16 as a potential biomarker. This cluster was identified in the revised analysis and is upregulated if *P. aeruginosa* is the primary colonizer (Figure 2, 4 & 5). Importantly, in response to 2) above, the statistical presentation has been thoroughly revised. The revised analysis shows that this cluster, among others, is significantly differentially abundant after multiple hypothesis testing corrections with the dataset size presented in our study. Hence, these data provide proof of principle demonstrating the application of the methodology in infected wounds. These changes have been incorporated into the main figures alongside several new supplemental figures and sections. We have also changed the title to "Peptide clusters reveal proteolytic signatures and biomarkers in mass spectrometry data" to emphasize the conceptual nature of our work and made changes in the text in all sections of the manuscript.

Version 1:

Reviewer comments:

Reviewer #1

(Remarks to the Author)

Reviewer #2

(Remarks to the Author)

The authors have made a substantial revision and addressed most of the questions pointed out by this reviewer. This reviewer is impressed by that the author summarized the amendments in clarity, utility, and reproducibility. Importantly, the authors have compared their clustering approach with ones previously published. However, there are still two weaknesses in the current version of the manuscript.

1. In response to the first major issue pointed out by this reviewer, the authors have made a comparison with traditional peptidomic approaches and also added the last section in the Results (at the bottom of Page 12 in the revised manuscript). However, the statement is too brief to show what essential information they can obtain by using the current clustering-based strategy, and what is its advantage in comparison to the traditional one. This is very important to showcase the technical innovation and to demonstrate the efficiency and utility of the newly-developed strategy. It would be a better structured manuscript that the data of comparison with traditional approach is moved to the first section of Results. In addition, the section of Abstract lacks key results/information to support the conclusion on the benefit of the new strategy, and the last paragraph of Introduction also lack such information.

2. In response to the third major issue pointed out by this reviewer, the authors said “We still concur with the assessment that our study primarily demonstrates the feasibility of our new strategy, rather than the identification of specific clinically validated disease markers” in the rebuttal letter. Therefore, it is improper to use “biomarker” in the title and the main manuscript.

Just, occasionally, a typo was found. If Figure 2b and 2d share one annotation bar, there are two duplicated “sample type”.

Reviewer #3

(Remarks to the Author)

Hartman and colleagues submitted a revised manuscript entitled “Peptide clusters reveal proteolytic signatures and biomarkers in mass spectrometry data” for publication. The authors have replied to my comments, but not always in the way I was hoping for. While the authors obviously have put in quite some efforts and the manuscript has been substantially improved, in the reviewers opinion it still has major shortcomings in key aspects.

The main aim of the publication is explained now in more detail: presenting an algorithm that enables clustering and dimensional reduction of peptidomics data. However, I am not able to detect an evident benefit in comparison to the state-of-the-art.

The authors use the term biomarker frequently, but it is unclear what the biomarker specifically reflects (please have a look at the definition of biomarker).

The problem of the missing values and the study being of very low power remains unchanged.

Overall an algorithm for dimensionality reduction of peptidomics data by clustering peptides is presented. If the application of this algorithm is in fact beneficial or if it may even result in loss of relevant information is unclear to this reviewer, as a result of the very limited sample size.

Reviewer #4

(Remarks to the Author)

Author Rebuttal letter:

General comments:

We thank the reviewers for another good round of reviews and constructive comments. We have now made changes to the manuscript to address the comments.

The main amendments are:

1. Moving and extending the comparison analysis on the dataset of type 1 diabetics to the main manuscript to highlight the technical utility of the clustering method.
2. Changing the usage of "biomarker" to fit the scope of the manuscript.
3. Further demonstrating the robustness of our data and its adequacy in the conducted analysis.

Beyond these, we have made minor changes to the main text that improve the flow of the manuscript. Point-by-point responses can be found below.

Reviewer #2 (Remarks to the Author):

The authors have made a substantial revision and addressed most of the questions pointed out by this reviewer. This reviewer is impressed by that the author summarized the amendments in clarity, utility, and reproducibility. Importantly, the authors have compared their clustering approach with ones previously published. However, there are still two weaknesses in the current version of the manuscript.

1. In response to the first major issue pointed out by this reviewer, the authors have made a comparison with traditional peptidomic approaches and also added the last section in the Results (at the bottom of Page 12 in the revised manuscript). However, the statement is too brief to show what essential information they can obtain by using the current clustering-based strategy, and what is its advantage in comparison to the traditional one. This is very important to showcase the technical innovation and to demonstrate the efficiency and utility of the newly-developed strategy. It would be a better structured manuscript that the data of comparison with traditional approach is moved to the first section of Results. In addition, the section of Abstract lacks key results/information to support the conclusion on the benefit of the new strategy, and the last paragraph of Introduction also lack such information.

We agree that the technical innovation of the clustering strategy needs to be highlighted in the main manuscript. We have therefore expanded on the previously included section and included a revised version of the figure as Fig. 7, comparing a peptide-centric and our cluster-centric approach using the dataset from Van et al. previously present in the supplement section. Importantly, the new panels show that the clustered data contains proportionally fewer missing values among the most discriminating features and that clustered data outperforms the traditional peptide-centric data during the classification of type 1 diabetes (Fig. 7h and 7i below). While we agree that this section has an important place in the manuscript, we didn't want to break the flow of the infection-based study by inserting the analysis of diabetes urine. We have therefore decided to keep this section towards the end of the results section. We have added statements about the utility in the introduction as follows on page 2.

“Lastly, we demonstrate its utility and generalizability on data from patients with type 1 diabetes, where we show that the clustering results in more informative features with fewer missing values which increases the classification accuracy.”

We have also added a statement in the abstract:

“... This revealed signature phenotype-specific peptide regions and proteolytic activity at the earliest stages of bacterial colonization. We validated the method on the urinary peptidome of type 1 diabetics which revealed potential subgroups and improved classification accuracy.”

The added section to the results is pasted below:

“To evaluate the efficacy and generalization of our computational methodology, we applied it to a publicly available dataset from a study by Van et al. [Van2020], downloaded from ProteomeXchange. This dataset comprises 30 urine samples analyzed using MS/MS peptidomic methodology, including 15 samples from patients with type 1 diabetes and 15 control samples. We re-created part of the analysis (Fig. 2e, f in [Van2020]), as shown in Fig. 7a, b, and c. P-values were calculated using linear regression and corrected with the Benjamini-Hochberg method for multiple hypothesis testing.

We then compared these results with our methodology (Fig. 7d). Clustering was performed with a cutoff of 4 and a resolution of 0.8, consistent with our other analyses. Clusters were quantified using the top 3 peptides with the highest DE scores. Our findings revealed that the uromodulin (UMOD) peptides identified by the classical approach are from the same cluster (UMOD 458-474). Additionally, we identified two new clusters (LMAN2 161-177 and SECTM1 118-139) that passed the Q-value cutoff of 0.05 (Fig. 7e). Interestingly, none of the peptides from these proteins were included in the top 15 peptides identified with a peptide-centric approach. Out of 43 clusters passing the p-value cutoff of 0.05, the top 20 clusters are visualized in Fig. 7f for clarity. The clustering approach revealed two apparent subtypes of type 1 diabetes that were not apparent using the traditional peptide-centric approach. Further analysis using UMAP projected the subsetted feature matrices to two dimensions, showing clear separation by sample type in both peptide-centric and cluster-centric workflows (Fig. 7g).

To illustrate the improvement in feature quality through clustering, we performed an iterative scheme to evaluate the number of missing values and the machine learning-based predictive power of progressively larger feature matrices. The original feature matrices were subsetted to include the top N features sorted by p-value, beginning with N=3. At each iteration, a feature was added. The number of missing values was plotted against the normalized number of features included in the feature matrices in Fig. 7h. The clustered data contains a smaller proportion of missing values among the most discriminating features compared to the peptide-centric data since these clusters utilize quantification information from a combined set of peptides. Secondly, we evaluated the area under the receiver operating characteristic curve (AUROC) across progressively larger feature matrices. 10 logistic regression classifiers were trained on different stratified subsets of half the dataset, and the AUROC was evaluated on the remaining halves. This iterative process continued until the full dataset size was reached. The resulting AUROCs across the feature space are shown in Fig. 7i. The AUROC of the clustered data was consistently higher than the peptide-centric data. This demonstrates that clustering combines peptides with missing information into more informative features.

[Redacted]

Fig. 7 Comparing a peptide-centric workflow with a cluster-centric workflow on a publicly available dataset. a Workflow re-creation from Van et al., filtering peptides present in all samples to yield a final dataset of 149 peptides. b Volcano plot corresponding to Fig. 2e in Van et al., displaying Q-values on the y-axis and highlighting 5 UMOD peptides. c Clustermap replicating Fig. 2f in Van et al. d Outline of the peptide cluster workflow. Clusters were included if present in at least half the dataset, resulting in 315 clusters that were quantified by the top 3 most differentially abundant peptides. e Volcano plot of quantified clusters, annotating clusters with Q-value < 0.05. f Clustermap of the top 20 clusters by p-value, highlighting two subgroups identified by hierarchical clustering. g UMAP projections of subsetted peptide and cluster matrices from c and f, highlighting subgroups identified in the cluster-centric UMAP. h Normalized number of missing values plotted against the normalized number of features in iteratively larger feature matrices. i AUROC versus the number of included features, comparing logistic regression classifiers trained on iteratively larger feature matrices for both peptide-centric and cluster-centric workflows. 10 classifiers were trained for each feature addition. All results are shown as faded lines and the non-faded lines represent the means.

2. In response to the third major issue pointed out by this reviewer, the authors said "We still concur with the assessment that our study primarily demonstrates the feasibility of our new strategy, rather than the identification of specific clinically validated disease markers" in the rebuttal letter. Therefore, it is improper to use "biomarker" in the title and the main manuscript.

We agree and have changed the title to: "Peptide clustering enhances large-scale analyses and reveals proteolytic signatures in mass spectrometry data". We have also removed the term "biomarkers" from the result section.

Just, occasionally, a typo was found. If Figure 2b and 2d share one annotation bar, there are two duplicated "sample type".

This has been corrected.

Reviewer #3 (Remarks to the Author):

Hartman and colleagues submitted a revised manuscript entitled "Peptide clusters reveal proteolytic signatures and biomarkers in mass spectrometry data" for publication. The

authors have replied to my comments, but not always in the way I was hoping for. While the authors obviously have put in quite some efforts and the manuscript has been substantially improved, in the reviewers opinion it still has major shortcomings in key aspects. The main aim of the publication is explained now in more detail: presenting an algorithm that enables clustering and dimensional reduction of peptidomics data. However, I am not able to detect an evident benefit in comparison to the state-of-the-art.

We are glad that the reviewer recognized the increased quality of the most recent version of the manuscript. We hope that the actions taken below convince the reviewer of the value of our manuscript and address the remaining concerns.

The authors use the term biomarker frequently, but it is unclear what the biomarker specifically reflects (please have a look at the definition of biomarker).

We agree with this remark and have altered the title to "Peptide clustering enhances large-scale analyses and reveals proteolytic signatures in mass spectrometry data" which better encapsulates the aim of the study. We have also removed the mention of biomarkers from the result section.

The problem of the missing values and the study being of very low power remains unchanged.

We recognize the reviewer's concern regarding the impact of missing values and the size of our dataset. The actions below address these concerns and demonstrate that 1) The dataset size of our study is relatively large and adequate to draw the biological conclusions presented in the paper. 2) The number of missing values does not pose a problem during the analysis of the clustered data.

To address the concerns regarding sample size, we performed a meta-literature review and plotted the number of samples included in similar studies. This shows that the dataset included in our study is substantially larger than most previously published data sets.

[Redacted]

Dataset sizes of LC-MS/MS peptidomic studies. A meta-literature review was conducted by identifying the number of samples of applicable studies present in Madsen et al. (2022, Nat. Comm.) Fig. 1 and Foreman et al. (2021, J. Proteome. Res.) Table 1. The largest study was conducted by Madsen et al. in 2022. Ours is the 2nd largest by number of samples.

To demonstrate the robustness of our study, we calculated the statistical power (the probability of avoiding a Type 2 error) for the peptide clusters shown in Figure 2d. Specifically, we assessed whether our study can reliably detect differences in the quantities of these clusters between samples infected with *P. aeruginosa* and those infected with *S. aureus*. The null hypothesis is that there is no difference in cluster quantities between the two types of infections. Power represents the likelihood of rejecting this null hypothesis when a true difference exists.

We assume that the effect distributions are t-distributed because the log-transformed MS/MS intensities follow a Gaussian distribution. We calculated the power at a significance level of 0.05. The results showed high power for all highlighted peptide clusters, indicating that our study is well-equipped to detect true differences in cluster expression. This high power confirms that our sample size, effect size, and variability are sufficient to ensure the sensitivity of our test.

[Redacted]

To investigate how the missing values in our peptidomic clustering data affected the biological outcome, we bootstrapped and down-sampled our data to 50% of its original dimensionality and then reduced it further using UMAP. The results are shown below, alongside the non-bootstrapped UMAP. If the number of missing values and irrelevant features greatly exceeded the number of informative features, bootstrapping and down-sampling would likely result in a high proportion of non-informative features. In such a case, the UMAP projection would not consistently show similar clustering patterns across different bootstrap iterations. However, our test showed that the data still clusters by time point and sample type, even after down-sampling. This indicates that our data is robust to the down-sampling process and the presence of missing values. This figure and paragraph

has been included as Supplementary Notes 6 and Supplementary Fig. 8.

[Redacted]

UMAP projections of bootstrapped and down-sampled peptide clusters. The original feature matrix contains the scaled quantities of 743 peptide clusters. During the bootstrap, 371 clusters are picked randomly and a UMAP is fitted to the subsampled data.

We have also added a section on the replicability and robustness of our data in the main manuscript:

âDue to the high number of missing values in the unprocessed dataset, we assessed the replicability and robustness of our data. We conducted a blinded re-run of a subset of samples, which confirmed the technical reproducibility of our findings (see Supplementary Notes 4 and Supplementary Fig. 6). Additionally, we applied down-sampled bootstrapping to the peptide cluster feature matrix before performing dimensionality reduction with UMAP. This analysis showed that missing values had little impact at the peptide cluster level, as the UMAP projections still clustered by sample type and time point (Supplementary Notes 5 and Supplementary Fig. 7).â

Overall an algorithm for dimensionality reduction of peptidomics data by clustering peptides is presented. If the application of this algorithm is in fact beneficial or if it may even result in loss of relevant information is unclear to this reviewer, as a result of the very limited sample size.

Above we have demonstrated the sample size of our study is comparatively large and that the power is high. Further, we have shown that the data is robust by performing a bootstrapped down-sampling before dimensionality reduction. Previously, we have also demonstrated the technical replicability of our study by performing a blinded re-run of samples.

In the previous round of reviews, we demonstrated the utility of our method by analyzing a publicly available dataset. This highlighted additional clusters of peptides which passed a Q-value threshold that was overlooked in the original study, alongside a potential subtype of diabetes type 1 which wasnât apparent in the traditional peptide-centric workflow. We have now moved and expanded on this comparison, thereby adding a new figure as Fig. 7. Importantly, the new panels in this figure show that the clustered data contains proportionally fewer missing values among the most discriminating features and that clustered data outperforms the traditional peptide-centric data during the classification of type 1 diabetes (Fig. 7h and 7i below). It should also be noted that the run-time to perform the classifications for Fig. 7i took <20x lower time on the clustered dataset as a result of the dimensionality reduction (from approx. 20 min to a few seconds on an M2 Mac with 38 cores). For training, the time complexity of common models scales linearly with increased dimensionality, making the time complexity of the iterative process $O(\text{dimensions}^2)$, showcasing another substantial benefit to dimensionality-reduced data. The following text paragraph to the main manuscript:

âTo evaluate the efficacy and generalization of our computational methodology, we applied it to a publicly available dataset from a study by Van et al. [Van2020], downloaded from ProteomeXchange. This dataset comprises 30 urine samples analyzed using MS/MS peptidomic methodology, including 15 samples from patients with type 1 diabetes and 15 control samples. We re-created part of the analysis (Fig. 2e, f in [Van2020]), as shown in Fig. 7a, b, and c. P-values were calculated using linear regression and corrected with the Benjamini-Hochberg method for multiple hypothesis testing.

We then compared these results with our methodology (Fig. 7d). Clustering was performed with a cutoff of 4 and a resolution of 0.8, consistent with our other analyses. Clusters were quantified using the top 3 peptides with the highest DE scores. Our findings revealed that the uromodulin (UMOD) peptides identified by the classical approach are from the same cluster (UMOD 458-474). Additionally, we identified two new clusters (LMAN2 161-177 and SECTM1 118-139) that passed the Q-value cutoff of 0.05 (Fig. 7e). Interestingly, none of the peptides from these proteins were included in the top 15 peptides identified with a peptide-centric approach. Out of 43 clusters passing the p-value cutoff of 0.05, the top 20 clusters are visualized in Fig. 7f for clarity. The clustering approach revealed two apparent subtypes of type 1 diabetes that were not apparent using the traditional peptide-centric approach. Further analysis using UMAP projected the subsetted feature matrices to two dimensions, showing clear separation by sample type in both peptide-centric and cluster-centric workflows (Fig. 7g).

To illustrate the improvement in feature quality through clustering, we performed an iterative

scheme to evaluate the number of missing values and the machine learning-based predictive power of progressively larger feature matrices. The original feature matrices were subsetted to include the top N features sorted by p-value, beginning with N=3. At each iteration, a feature was added. The number of missing values was plotted against the normalized number of features included in the feature matrices in Fig. 7h. The clustered data contains a smaller proportion of missing values among the most discriminating features compared to the peptide-centric data since these clusters utilize quantification information from a combined set of peptides. Secondly, we evaluated the area under the receiver operating characteristic curve (AUROC) across progressively larger feature matrices. 10 logistic regression classifiers were trained on different stratified subsets of half the dataset, and the AUROC was evaluated on the remaining halves. This iterative process continued until the full dataset size was reached. The resulting AUROCs across the feature space are shown in Fig. 7i. The AUROC of the clustered data was consistently higher than the peptide-centric data. This demonstrates that clustering combines peptides with missing information into more informative features.

[Redacted]

Fig. 7 Comparing a peptide-centric workflow with a cluster-centric workflow on a publicly available dataset. a Workflow re-creation from Van et al., filtering peptides present in all samples to yield a final dataset of 149 peptides. b Volcano plot corresponding to Fig. 2e in Van et al., displaying Q-values on the y-axis and highlighting 5 UMOD peptides. c Clustermap replicating Fig. 2f in Van et al. d Outline of the peptide cluster workflow. Clusters were included if present in at least half the dataset, resulting in 315 clusters that were quantified by the top 3 most differentially abundant peptides. e Volcano plot of quantified clusters, annotating clusters with Q-value < 0.05. f Clustermap of the top 20 clusters by p-value, highlighting two subgroups identified by hierarchical clustering. g UMAP projections of subsetted peptide and cluster matrices from c and f, highlighting subgroups identified in the cluster-centric UMAP. h Normalized number of missing values plotted against the normalized number of features in iteratively larger feature matrices. i AUROC versus the number of included features, comparing logistic regression classifiers trained on iteratively larger feature matrices for both peptide-centric and cluster-centric workflows. 10 classifiers were trained for each feature addition. All results are shown as faded lines and the non-faded lines represent the means.

Additionally, we compared the proposed method to a traditional workflow on our data. This demonstrated the utility of using clusters by reducing the amount of redundant information into more informative features, decreasing the level of noise for cut-site predictions, reducing the dataset size and increasing interpretability.

With these amendments, we have demonstrated the replicability, robustness and utility of our study.
